# Transductive and Learning-Augmented Online Regression

## Abstract

Motivated by the predictable nature of real-life in data streams, we study online regression when the learner has access to predictions about future examples. In the extreme case, called transductive online learning, the sequence of examples is revealed to the learner before the game begins. For this setting, we fully characterize the minimax expected regret in terms of the fat-shattering dimension, establishing a separation between transductive online regression and (adversarial) online regression. Then, we generalize this setting by allowing for noisy or *imperfect* predictions about future examples. Using our results for the transductive online setting, we develop an online learner whose minimax expected regret matches the worst-case regret, improves smoothly with prediction quality, and significantly outperforms the worst-case regret when future example predictions are precise, achieving performance similar to the transductive online learner. This enables learnability for previously unlearnable classes under predictable examples, aligning with the broader learning-augmented model paradigm.

## 1 Introduction

Online learning is framed as a sequential game between a learner and an adversary. In each round, the learner first makes a prediction after which the adversary evaluates the learner's prediction, typically by producing a ground-truth label. In contrast to the classical batch learning setting, where one places distributional assumptions on the data-generating process, in online learning, we place no assumptions on the data-generating process, even allowing the adversary to be adaptive to the past actions of the learner. Due to its level of generality, online learning has received substantial attention over the years. While online learning literature is too vast to review comprehensively, we include a detailed discussion of the most relevant works in Appendix A.

In this work, we focus on online *regression*, where the learner's predictions are measured via a well-structured loss function $\ell(\cdot, \cdot)$, e.g., the $\ell_1$ loss $\ell(y, \widehat{y}) = |y - \widehat{y}|$. The online regression problem is formally defined as the following $T$-round interactions between the learner $\mathcal{A}$ and the adversary: In each round $t \in [T]$, the adversary picks a labeled example $(x_t, y_t)$ from $\mathcal{X} \times \mathcal{Y}$ and reveals the example $x_t$ to the learner. The learner then predicts $\widehat{y}_t$ based on historical observations $(x_1, y_1), \ldots, (x_{t-1}, y_{t-1})$ and the current example $x_t$. Finally, the adversary reveals the actual label $y_t$ to the learner, and the learner suffers loss $\ell(\widehat{y}_t, y_t)$. Given a function class $\mathcal{F} \subset \mathcal{Y}^{\mathcal{X}}$, the goal of the learner is to minimize the minimax expected regret

$$R_{\mathcal{A}}(T, \mathcal{F}) := \sup_{(x_1, \ldots, x_T) \subset \mathcal{X}} \sup_{(y_1, \ldots, y_T) \subset \mathcal{Y}} \left( \mathbb{E}_{\mathcal{A}} \left[ \sum_{t=1}^{T} \ell(\mathcal{A}_t, y_t) \right] - \inf_{f \in \mathcal{F}} \sum_{t=1}^{T} \ell(f(x_t), y_t) \right),$$

which is defined as the difference between the cumulative loss of the learner $\mathcal{A}$ and the cumulative loss of the best function in $\mathcal{F}$. We say that a function class $\mathcal{F}$ is online learnable if $\inf_{\mathcal{A}} R_{\mathcal{A}}(T, \mathcal{F}) = o(T)$, that is, there exists a learner who achieves average regret that is sublinear in $T$ for all possible choices of labeled examples given by the adversary.

Rakhlin et al. (2015a) showed that a combinatorial parameter called the sequential fat-shattering dimension fully characterizes learnability – a class is online learnable if and only if its sequential fat shattering dimension is finite. However, this result is discouraging given the restrictive nature of the sequential fat shattering dimension. For instance, even simple function classes, such as functions

with bounded variation in $[0, 1]$, have infinite sequential fat-shattering dimensions, which means that they are not online learnable. This challenge arises from worst-case scenarios, as the adversary is allowed to choose any labeled example sequences, potentially adapting to the learner's output. In practice, however, data streams often exhibit predictable patterns, so the worst-case assumption can be relaxed (Raman & Tewari, 2024). Given this intuition, we investigate online regression with various levels of prior knowledge about the sequence of examples $x_1, \ldots, x_T$. As motivation, consider the following example.

**Example 1.1.** *A smart building management system models energy consumption $y_t \in \mathbb{R}$ as a function of features $x_t \in \mathbb{R}^d$, such as temperature and occupancy count, to estimate daily energy usage. If predicted consumption significantly deviates from actual consumption $y_t$, either underestimation causes energy shortages or overestimation wastes resources, incurring a loss $\ell(y_t, \widehat{y}_t)$.*

*In practice, the system has prior knowledge of daily occupancy schedules and weather forecasts. However, it does not know exactly the energy consumption $y_t$ due to variable factors such as occupant behavior, equipment usage, or unexpected events. Thus, it predicts energy consumption based on its knowledge of the sequence of examples $x_1 \ldots x_T$, which forms an online regression problem.*

Real-world scenarios, like the example above, have inspired recent research on learning-augmented algorithms (Mitzenmacher & Vassilvitskii, 2020), which enhances the algorithm's performance using additional information about the problem instance given by machine-learned predictions. For example, machine-learned predictions have been utilized to achieve more efficient data structures (Mitzenmacher, 2018; Lin et al., 2022; Fu et al., 2025), algorithms with faster runtimes (Dinitz et al., 2021; Chen et al., 2022c; Davies et al., 2023), mechanisms with better accuracy-privacy trade-offs (Khodak et al., 2023), streaming algorithms with better accuracy-space tradeoffs (Hsu et al., 2019; Indyk et al., 2019; Jiang et al., 2020; Chen et al., 2022b;a; Li et al., 2023), and accuracy guarantees beyond NP hardness (Ergun et al., 2022; Nguyen et al., 2023; Karthik C. S. et al., 2025). A more detailed summary is deferred to Appendix A. Motivated by work on learning-augmented algorithms, in this paper, we study the following question:

*Can predictions about the future examples be used to get better-than-worst-case regret bounds for online regression?*

Rakhlin & Sridharan (2013) studied this question in the context of online linear optimization. Specifically if the sequence encountered by the learner is described well by a known "predictable process", their algorithms enjoy tighter bounds as compared to the typical worst case bounds. Additionally, their methods achieved the usual worst-case regret bounds if the sequence is not benign. More recently, Raman & Tewari (2024) studied this question in the context of online *classification*. Their proposed algorithm performs optimally when the predictions are nearly exact, while ensuring the worst-case guarantee. Furthermore, they characterize the expected number of mistakes as a function of the quality of predictions, interpolating between instance and worst-case optimality. In this work, we study this same question in the more general setting of online (non-parametric) *regression*. We demonstrate that learning-augmented algorithms achieve better minimax expected regret compared to the online learner under worst-case scenarios. We consider two settings where we apply the learning-augmented framework: the transductive online regression setting where the learner has full knowledge about the sequence of examples; and the online regression with predictions setting, where the learner has access to a Predictor for future examples.

**Transductive online learning.** In transductive online learning, initially introduced by Ben-David et al. (1997) and recently studied by Hanneke et al. (2024a), the entire sequence of examples $x_1, \ldots, x_T$ picked by the adversary is revealed to the learner before the game starts. In each round $t \in [T]$, the learner makes a prediction $\widehat{y}_t$ using the information from $(x_1, y_1), \ldots, (x_t, y_t), x_{t+1}, \ldots, x_T$. Lastly, the adversary reveals the actual label $y_t$ to the learner, and the learner suffers a loss. In many situations, however, we do not have full access to the examples $x_1, \ldots, x_T$. This motivates a generalization where the learner has access to potentially noisy predictions of future examples.

**Online regression with predictions.** In online learning with predictions (Raman & Tewari, 2024), the learner has access to a Predictor $\mathcal{P}$, which observes the past examples $x_1, \ldots, x_t$ and predicts future examples. In each round $t \in [T]$, the learner $\mathcal{A}$ queries the Predictor and receives potentially

noisy predictions $\widehat{x}_{t+1}, \ldots, \widehat{x}_T$. The learner $\mathcal{A}$ then makes a prediction $\widehat{y}_t$ based on the current example $x_t$, the predictions $\widehat{x_{t+1}}, \ldots, \widehat{x}_t$, and the previous labeled-examples $(x_1, y_1), \ldots (x_{t-1}, y_{t-1})$. We allow the Predictor to be adaptive, which means that can change its predictions about future examples based on the current example $x_t$. We quantify the performance of a Predictor $\mathcal{P}$ using two metrics: the zero-one metric that counts the number of times its prediction of the next example is wrong and the $\varepsilon$-ball metric that counts the number of times its prediction of the next example is sufficiently far away from the true next example with respect to some metric of interest.

## 1.1 OUR RESULTS

In this work, we seek to understand how the regret scales as a function of the quality of the predictions. In particular, under what circumstances can we do better than the worst-case regret? Motivated by this question, we provide the following result for transductive online regression.

**Theorem 1.2** (Transductive online regression, informal statement for Theorem 3.1). *A function class $\mathcal{F} \subseteq [0,1]^{\mathcal{X}}$ is transductive online learnable under the $\ell_1$-loss if and only if its fat-shattering dimension is finite.*

This result establishes a separation between transductive online regression and online regression for function classes with finite fat-shattering dimensions but infinite sequential fat-shattering dimensions, e.g., the class of functions with bounded variations. Specifically, any function classes with infinite sequential fat-shattering dimension is not online learnable due to the lower bound in Rakhlin et al. (2015a), but can be transductive online learnable if it has finite fat-shattering dimension. As a corollary of Theorem 1.2, we get that while the class of functions with $V$-bounded variations is not online learning, it is transductive online learnable with minimax regret scaling like $\tilde{O}(\sqrt{VT})$.

Motivated by scenarios where having full access to $x_1, \ldots, x_T$ is unrealistic, our second result studies online regression with predictions, where instead of having access to $x_1, \ldots, x_T$, the learner has black-box access to a Predictor $\mathcal{P}$ and a transductive online learner $\mathcal{B}$. In this more general setting, we give an online transductive learner whose minimax expected regret can be written a function of the mistake-bound of $\mathcal{P}$ and interpolates between instance and worst-case optimality depending on the quality of $\mathcal{P}$'s predictions, (see Theorem 4.1 for a precise statement). Our learning algorithm is *consistent* in that it has the same minimax expected regret as $\mathcal{B}$ when the predictions are exact, and is *robust* in that it never has worse regret than the minimax optimal regret in the fully adversarial online learning model.

We measure the mistake-bound of $\mathcal{P}$ with respect to two different metrics. The first is the *zero-one metric*, which measures the expected number of incorrect predictions $\widehat{x}_t \neq x_t$. As examples are often noisy in real-world applications, perfectly predicting the next example is unlikely. As a result, our second metric is the *$\varepsilon$-ball metric*, which measures the expected number of predictions that are outside of the $\varepsilon$-ball of the actual example. To get a sense of how the minimax expected regret of our learner scales with the mistake-bound of our Predictor, Corollary 1.3 provides upper bounds on the minimax expected regret of our online learner for both mistake-bound guarantees for the Predictor. Here, we omit the polylogarithmic factors in $T$.

**Corollary 1.3** (Informal statement for online regression with predictions). *Given a Predictor $\mathcal{P}$ and a transductive online learner $\mathcal{B}$ with minimax expected regret $R_{\mathcal{B}}^{\mathbf{tr}}(T, \mathcal{F})$:*

- *If $\mathcal{P}$ makes $\mathcal{O}(T^p)$ mistakes under the zero-one metric, then for any function class $\mathcal{F} \subseteq [0,1]^{\mathcal{X}}$, there is an online learner $\mathcal{A}$ whose minimax expected regret under the $\ell_1$ loss is at most*

$$\mathcal{O}(T^p) R_{\mathcal{B}}^{\mathbf{tr}}(T^{1-p}, \mathcal{F}) + \sqrt{T \log^2 T}.$$

- *If $\mathcal{P}$ makes $M_{\mathcal{P}}(\varepsilon, x_{1:T}) = \mathcal{O}\left(\frac{T^p}{\varepsilon^q}\right)$ mistakes under the $\varepsilon$-ball metric, then for any $L_{\mathbf{hyp}}$-Lipschitz function class $\mathcal{F} \subseteq [0,1]^{\mathcal{X}}$, there is an online learner $\mathcal{A}$ whose minimax expected regret under the $\ell_1$ loss is at most*

$$\inf_{\varepsilon > 0} \left\{ \mathcal{O}\left(\frac{T^p}{\varepsilon^q}\right) R_{\mathcal{B}}^{\mathbf{tr}}(\varepsilon^q T^{1-p}, \mathcal{F}) + \varepsilon L_{\mathbf{los}} L_{\mathbf{hyp}} \cdot T + \sqrt{T \log^2 T} \right\}.$$

Corollary 1.3 is a combination of Theorem 4.2 (minimax expected regret under the zero-one metric) and Theorem 4.4 (minimax expected regret under the $\varepsilon$-ball metric). As a concrete example, we

show that the function class with bounded variation is online learnable if the sequence of examples is predictable under the zero-one metric, that is, the number of mistakes of the Predictor grows sublinearly with the time horizon (see Corollary 4.5). In addition, we identify a subclass of functions with bounded variation and a large Lipschitz constant (see Definition 4.6), such that it is not online learnable under the worst case, but online learnable given a Predictor with small error rate under the $\varepsilon$-ball metric. These results establish a separation between online regression with predictions and online regression for various function classes. We note that our results answer the open question in Section 4 of Raman & Tewari (2024) about the learnability of online regression under general measures of predictability.

## 2 PRELIMINARIES

Let $\mathcal{X}$ denote the example space, let $\mathcal{Y} = [0, 1]$ denote the label space, let $\mathcal{F} \subset \mathcal{Y}^{\mathcal{X}}$ denote the function class, and let $\ell(\widehat{y}, y) = |\widehat{y} - y|$ be the $\ell_1$-loss function (we consider a more general notion of convex and $L_{\mathbf{los}}$-Lipschitz loss function in the Appendix). Let $T$ denote the length of the sequence of examples. Let $z_{1:T}$ denote the sequence of items $z_1, \ldots, z_T$. Let $\tilde{\mathcal{O}}(f) = f \cdot \mathrm{polylog}(f)$. Given some event $\mathcal{E}$, let $\mathbf{1}_{\mathcal{E}}$ be the indicator function, which is 1 if $\mathcal{E}$ holds, and 0 otherwise. Next, we introduce the formal definitions of the minimax expected regret and selected complexity measures. The complete version is deferred to Appendix B.

### 2.1 ONLINE LEARNING

In the standard online learning setting, the game proceeds over $T$ rounds of interactions between the learner $\mathcal{A}$ and the adversary: In each round $t \in [T]$, the adversary picks a labeled example $(x_t, y_t) \in \mathcal{X} \times \mathcal{Y}$ and reveals $x_t$ to the learner. The learner then produces a prediction $\widehat{y}_t$ and receives the true label $y_t$. Given a function class $\mathcal{F} \subset \mathcal{Y}^{\mathcal{X}}$, the learner aims to minimize the expected regret,

$$R_{\mathcal{A}}^{\mathbf{ol}}(T, \mathcal{F}) := \sup_{x_{1:T} \in \mathcal{X}^T} \sup_{y_{1:T} \in \mathcal{Y}^T} \left( \mathbb{E}\left[ \sum_{t=1}^{T} |\mathcal{A}(x_t) - y_t| \right] - \inf_{h \in \mathcal{F}} \sum_{t=1}^{T} |h(x_t) - y_t| \right),$$

where the expectation is over randomness of the learner. We say that a function class $\mathcal{F}$ is online learnable if $\inf_{\mathcal{A}} R_{\mathcal{A}}^{\mathbf{ol}}(T, \mathcal{F}) = o(T)$. Rakhlin et al. (2015a) showed that, in the online setting, the expected regret is controlled by the sequential fat-shattering dimension $\mathrm{fat}_{\alpha}^{\mathbf{seq}}(\mathcal{F})$. We defer the formal definition and statement to Appendix B.

### 2.2 TRANSDUCTIVE ONLINE LEARNING.

In transductive online learning, unlike online learning, the adversary first reveals the entire sequence of unlabeled examples $x_1, \ldots, x_T$ to the learner at the beginning. The interaction then proceeds in $T$ rounds: In each round, the learner predicts a label $\widehat{y}_t$ for the current example $x_t$, using information from the past labeled examples $(x_1, y_1), \ldots, (x_t, y_t)$ and the future unlabeled examples $x_{t+1}, \ldots, x_T$. After the prediction, the adversary reveals the true label $y_t$. For a transductive online learner $\mathcal{B}$, we define its minimax expected regret as

$$R_{\mathcal{B}}^{\mathbf{tr}}(T, \mathcal{F}) := \sup_{x_{1:T} \in \mathcal{X}^T} \sup_{y_{1:T} \in \mathcal{Y}^T} \left( \mathbb{E}\left[ \sum_{t=1}^{T} |\mathcal{B}_{x_{1:T}}(x_t) - y_t| \right] - \inf_{h \in \mathcal{F}} \sum_{t=1}^{T} |h(x_t) - y_t| \right),$$

where again the expectation is over the randomness of the learner. We say that a function class $\mathcal{F}$ is transductive online learnable if $\inf_{\mathcal{B}} R_{\mathcal{B}}^{\mathbf{ol}}(T, \mathcal{F}) = o(T)$. In this paper, we characterize the learnability of transductive online regression by the fat-shattering dimension, a scale-sensitive version of the Vapnik-Chervonenkis (VC) dimension (Vapnik & Chervonenkis, 1971), which is also used to characterize PAC learnability (Alon et al., 1997).

**Definition 2.1** (Fat-shattering dimension, (Alon et al., 1997)). *A sequence $x = x_{1:T}$ is defined to be $\alpha$-shattered by $\mathcal{F}$ if there exists a sequence of real numbers $y = y_{1:T}$ such that for each binary string $\sigma \in \{-1, 1\}^T$, there is a function $f \in \mathcal{F}$ that satisfies*

$$\forall t \in [T], \quad \sigma_t \cdot (f(x_t) - y_t) \geq \alpha/2.$$

*Here, the sequence $y$ is called the witness of shattering. Then, the fat-shattering dimension $\mathrm{fat}_{\alpha}(\mathcal{F})$ is defined as the largest $T$ such that $\mathcal{F}$ $\alpha$-shatters a sequence $x \subset \mathcal{X}$ of length $T$. In addition, $\mathrm{fat}_{\alpha}(\mathcal{F}) = \infty$ if for every finite $T$, there is a sequence of length $T$ that is $\alpha$-shattered by $\mathcal{F}$.*

## 2.3 ONLINE LEARNING WITH PREDICTIONS

Since in practice, the sequence of examples often follows predictable patterns, we study the setting of online learning with predictions (Raman & Tewari, 2024). Here, the learner has access to a Predictor $\mathcal{P}$. $\mathcal{P}$ which predicts the sequence of examples $x_1, \ldots x_T$ adaptively: at each round $t \in [T]$, the adversary reveals the example $x_t$ to $\mathcal{P}$, then $\mathcal{P}$ reports the predictions of the *entire* sequence $\widehat{x}_1, \ldots \widehat{x}_T$ based on the past examples $x_1, \ldots, x_t$. We assume that $\mathcal{P}$ predicts the entire sequence for notational convenience, since it can set $\widehat{x}_c = x_c$ for each $c \in [t]$. We denote the prediction of $x_{t'}$ at time $t$ as $\mathcal{P}(x_{1:t})_{t'}$. Given the predictions, the learner predicts a label $\widehat{y}_t$ for the current example $x_t$, using information from the past labeled examples $(x_1, y_1), \ldots, (x_t, y_t)$ and the predictions of future examples $\widehat{x}_{t+1}, \ldots, \widehat{x}_T$. Here, we consider the standard adversarial setting in online learning, where the sequence of examples is *not* revealed to the learner in advance, and similarly, we measure the minimax expected regret by $R_{\mathcal{A}}^{\mathsf{ol}}(T, \mathcal{F})$. In this paper, we show that given a Predictor $\mathcal{P}$ with specific mistake-bounds, the minimax expected regret is characterized by the fat-shattering dimension but not the sequential version, and thus separate this setting from the standard online learning setting.

## 3 MINIMAX EXPECTED REGRET FOR TRANSDUCTIVE ONLINE REGRESSION

In this section, we give near-matching upper and lower bounds on the minimax expected transductive regret in terms of the fat-shattering dimension. Our main result is Theorem 3.1. Like the fully adversarial online setting, our proof is non-constructive and relies on minimax arguments. To that end, we also give an explicit a transductive online learner with sub-optimal regret based on the multiplicative weights algorithm (MWA). We defer the discussion of this learner to Appendix C.1.

**Theorem 3.1** (Minimax Expected Regret for Transductive Online Regression)**.** *For any function class $\mathcal{F} \subset [0,1]^{\mathcal{X}}$ and $\ell_1$-loss, we have the following bounds for the minimax expected regret of transductive online learning:*

$$\text{Lower bound}: \inf_{\mathcal{A}} R_{\mathcal{A}}^{\mathsf{tr}}(T, \mathcal{F}) \geq \sup_{\alpha} \left( \frac{\alpha}{4} \cdot \sqrt{T \cdot \min\{\mathrm{fat}_\alpha(\mathcal{F}), T\}} \right).$$

$$\text{Upper bound}: \inf_{\mathcal{A}} R_{\mathcal{A}}^{\mathsf{tr}}(T, \mathcal{F}) \leq 2T \cdot \inf_{\alpha \geq 0} \left( 4\alpha + \frac{12}{\sqrt{T}} \int_\alpha^1 \sqrt{\mathrm{fat}_{\beta/4}(\mathcal{F}) \cdot c \log^2 \frac{T}{\beta}} \mathrm{d}\beta \right).$$

Our approach to prove an upper bound on the minimax expected regret extends from the randomized learner framework in (Rakhlin et al., 2015a): Suppose that $\mathcal{Q}$ is a weakly compact set of probability measures on $\mathcal{F}$, then in each round $t$, $\mathcal{A}$ selects a probability measure $q_t \in \mathcal{Q}$ and outputs $\mathcal{A}_t = f_t(x_t)$, where $f_t \sim q_t$. We write $\mathcal{A}_t \sim q_t$ in the following for simplicity. Then, the minimax expected regret is represented as a minimax value, which encrypts the interaction of the learner and the adversary in each round:

$$\inf_{\mathcal{A}} R_{\mathcal{A}}^{\mathsf{tr}}(T, \mathcal{F}) = \sup_{x_{1:T}} \inf_{q_1 \in \mathcal{Q}} \sup_{y_1 \in \mathcal{Y}} \mathbb{E}_{\mathcal{A}_1 \sim q_1} \cdots \inf_{q_T \in \mathcal{Q}} \sup_{y_T \in \mathcal{Y}} \mathbb{E}_{\mathcal{A}_T \sim q_T} \left[ \sum_{t=1}^T \ell(\mathcal{A}_t, y_t) - \inf_{f \in \mathcal{F}} \sum_{t=1}^T \ell(f(x_t), y_t) \right].$$

Our key observation is that, since we have full access to $x_{1:T}$, $\sup_{x_{1:T}}$ is written in front of the minimax value in the above formula. Thus, we ultimately get an upper bound by the Rademacher complexity, instead of the sequential Rademacher complexity in Rakhlin et al. (2015a). Then, applying the entropy bound in Theorem B.2 gives an upper bound in terms of the fat-shattering dimension. We state the formal results in the following theorem.

For the lower bound, our proof is inspired by the hard instance for transductive online binary classification (Hanneke et al., 2023b), where they constructed the sequence of example by $k$ copies of sequence $x_1^*, \ldots, x_d^*$ that is VC-shattered by the function class and then apply the anti-concentration property of Rademacher variables. In our setting, we prove an equivalence expression of the expected regret formula to apply the definition of the fat-shattering dimension. The formal proofs are defered to Appendix C.2 (upper bound) and Appendix C.3 (lower bound).

We note that the above upper bound is in terms of the fat-shattering dimension of $\mathcal{F}$, as opposed to the sequential fat-shattering dimension in the online setting. Thus, our rate is better since many function classes have a finite fat-shattering dimension but an infinite sequential fat-shattering dimension, e.g., the class of functions with bounded variation.

To get a better sense of how the upper bound in Theorem 3.1 scales with $T$, Corollary 3.2 instantiates Theorem 3.1 with three concrete function classes: $L$-Lipschitz functions, $k$-fold aggregations, and functions with $V$-bounded variation. We present formal definitions of these classes and the proof of Corollary 3.2 in Appendix C.4.

**Corollary 3.2** (Examples). *Given a metric space $\mathcal{X} \subset \mathbb{R}^n$ of finite diameter $\mathrm{diam}(\mathcal{X})$, a function class $\mathcal{F} \subset [0,1]^{\mathcal{X}}$, and the $\ell_1$-loss function, the following statements are true:*

- *Lipschitz functions: If all functions in $\mathcal{F}$ are $L_{\mathbf{hyp}}$-Lipschitz, we have*

$$
R^{\mathbf{tr}}(T, \mathcal{F}) = \begin{cases} \tilde{\mathcal{O}}\left(\sqrt{L_{\mathbf{hyp}}} \cdot \sqrt{T}\right), & n = 1 \\ \tilde{\mathcal{O}}\left(L_{\mathbf{hyp}} \cdot \sqrt{T}\right), & n = 2 \\ \tilde{\mathcal{O}}\left(L_{\mathbf{hyp}} \cdot T^{\frac{n}{n+1}}\right), & n \geq 3 \end{cases} \cdot
$$

- *$k$-fold aggregations: If $\mathcal{F}$ is defined by an aggregation mapping $G(\mathcal{F}_1, \ldots, \mathcal{F}_k)$, where $G$ commutes with shifts, and $\mathcal{F}_1, \ldots, \mathcal{F}_k \subset [0,1]^{\mathcal{X}}$ are $L_{\mathbf{hyp}}$-Lipschitz function classes, then we have*

$$
R^{\mathbf{tr}}(T, \mathcal{F}) = \begin{cases} \tilde{\mathcal{O}}\left(\sqrt{k}\sqrt{L_{\mathbf{hyp}}} \cdot \sqrt{T}\right), & n = 1 \\ \tilde{\mathcal{O}}\left(\sqrt{k}L_{\mathbf{hyp}} \cdot \sqrt{T}\right), & n = 2 \\ \tilde{\mathcal{O}}\left(\sqrt{k}L_{\mathbf{hyp}} \cdot T^{\frac{n}{n+1}}\right), & n \geq 3 \end{cases} \cdot
$$

- *Functions with bounded variation: If $\mathcal{F}$ is the set of all functions $f : [0,1] \to [0,1]$ with total variation of at most $V$, then we have $R^{\mathbf{tr}}(T, \mathcal{F}) = \tilde{\mathcal{O}}\left(\sqrt{VT}\right)$.*

The upper bounds in our result demonstrate that all the function classes are transductive online learnable under $\ell_1$-loss. Note that for Lipschitz classes, the above result does not essentially give us a better rate for transductive online learning, since the sequential fat-shattering dimension of Lipschitz classes are roughly equivalent to their fat-shattering dimension (Rakhlin et al., 2015b). In contrast, the class of $k$-fold aggregations and the class of functions with bounded variation both have infinite sequential fat-shattering dimension, so they are not online learnable in the worst case. For these classes, our results establishes a separation between online learning and transductive online learning.

## 4 Minimax Expected Regret Bounds for Online Regression with Predictions

In this section, we consider the more general online regression with predictions setting. We construct learning algorithms given black-box access to a Predictor $\mathcal{P}$ and a transductive online learner $\mathcal{B}$. We compute the minimax expected regret in terms of the quality of $\mathcal{P}$ and $\mathcal{B}$.

We measure the performance of the Predictor $\mathcal{P}$ using two different metrics:

(1) Zero-one metric $M_{\mathcal{P}}(x_{1:T})$ which measures the expected number of incorrect predictions $\widehat{x}_t \neq x_t$, i.e., $M_{\mathcal{P}}(x_{1:T}) := \mathbb{E}\left[\sum_{t=2}^T \mathbf{1}_{\mathcal{P}(x_{1:t-1})_t \neq x_t}\right]$.

(2) $\varepsilon$-ball metric $M_{\mathcal{P}}(\varepsilon, x_{1:T})$ which measures the expected number of times that the prediction is outside the $\varepsilon$-ball: $\mathrm{d}(\widehat{x}_t, x_t) \geq \varepsilon$, where $\mathrm{d}$ is the metric on $\mathcal{X}$, i.e., $M_{\mathcal{P}}(\varepsilon, x_{1:T}) := \mathbb{E}\left[\sum_{t=2}^T \mathbf{1}_{\mathrm{d}(\mathcal{P}(x_{1:t-1})_t, x_t) \geq \varepsilon}\right]$.

Our main result in this section is Theorem Theorem 4.1, which bounds the minimax expected regret in terms of the mistake-bound of the Predictor and the regret of the transductive online learner. The proof is in Appendix D and is constructive.

**Theorem 4.1** (Online regression with predictions). *For every function class $\mathcal{F} \subset \mathcal{Y}^{\mathcal{X}}$, Predictor $\mathcal{P}$, transductive online learner $\mathcal{B}$ and $\ell_1$-loss, there exists an online learner $\mathcal{A}$ such that for every data*

*stream $(x_1, y_1), \ldots, (x_T, y_T)$ given by the adversary, the minimax expected regret of $\mathcal{A}$ is at most*

$$\min\left\{ \underbrace{R^{\mathbf{ol}}(T, \mathcal{F})}_{(a)}, \underbrace{2(M_{\mathcal{P}}(x_{1:T}) + 1)R_{\mathcal{B}}\left(\frac{T}{M_{\mathcal{P}}(x_{1:T}) + 1} + 1, \mathcal{F}\right)}_{(b)} \right\} + 2\sqrt{T \log T}$$

*In addition, if the functions in the class are $L_{\mathbf{hyp}}$-Lipschitz, the minimax expected regret is also upper bounded by*

$$\underbrace{2(M_{\mathcal{P}}(\varepsilon, x_{1:T}) + 1)R_{\mathcal{B}}\left(\frac{T}{M_{\mathcal{P}}(\varepsilon, x_{1:T}) + 1} + 1, \mathcal{F}\right) + \varepsilon L_{\mathbf{hyp}} \cdot T}_{(c)} + 2\sqrt{T \log T}.$$

We highlight the implications of each error bound. Firstly, the expected error of our algorithm is at most the worst-case error bound $(a)$ in the online setting. Secondly, the bound $(b)$ interpolates between the worst-case minimax expected regret and the tranductive online minimax expected regret as a function of $M_{\mathcal{P}}(x_{1:T})$, and when the Predictor is exact, i.e., $M_{\mathcal{P}}(x_{1:T}) = 0$, we get the same error bound as in the transductive online setting up to constants. Lastly, the bound $(c)$ relaxes the bounds of the Predictor to the more general $\varepsilon$-ball metric. Given a Lipschitz function class, our online learner has an expected regret sublinear in $T$ if the Predictor has sufficiently small error scales in terms of $\varepsilon$ and $T$. In Section 4.1, we explicitly compute the rates in $(b)$ and $(c)$, giving the sufficient conditions on the Predictors to achieve online learnability. Furthermore, we identify a class of functions with bounded variations that are not online learnable in the worst case but online learnable given desirable Predictors, and we highlight that existing Predictors suffice for the sequence of examples $x_{1:T}$ defined by a linear dynamical system.

**Overview of the algorithms.** We give an overview of our online learners. Under the zero-one metric $M_{\mathcal{P}}(x_{1:T})$, our online learner mainly follows from the construction in (Raman & Tewari, 2024) for online classification. We suppose that the prediction is incorrect at times $t_1, \ldots t_c \in [T]$ and correct at all other times. In the first algorithm, whenever $\mathcal{P}$ makes a mistake, the learner queries its new sequence of predictions and starts a transductive online learner $\mathcal{B}$ to predict $\widehat{y}_t$ until the next time $\mathcal{P}$ makes a mistake. This gives an error rate of roughly $M_{\mathcal{P}}(x_{1:T}) \cdot R_{\mathcal{B}}^{\mathbf{tr}}(T, \mathcal{F})$.

However, a crucial drawback of this algorithm is that, when $M_{\mathcal{P}}(x_{1:T})$ is large (e.g., $\Omega(\sqrt{T})$), the upper bound is suboptimal. To overcome this, in the second algorithm, we partition the time duration $[T]$ to $c$ equi-distant intervals and run a fresh copy of the first algorithm for each interval. Then we run MWA using experts with all $c \in [T - 1]$ as inputs. We show that the minimax expected regret for each expert with input $c$ is roughly $(M_{\mathcal{P}}(x_{1:T}) + c) \cdot R_{\mathcal{B}}^{\mathbf{tr}}\left(\frac{T}{c}, \mathcal{F}\right)$, which is $2M_{\mathcal{P}}(x_{1:T}) \cdot \bar{R}_{\mathcal{B}}^{\mathbf{tr}}\left(\frac{T}{M_{\mathcal{P}}(x_{1:T})}, \mathcal{F}\right)$ for the expert with $c = M_{\mathcal{P}}(x_{1:T})$, thus MWA gives an minimax expected regret having $M_{\mathcal{P}}(x_{1:T})$ as an interpolation factor and only loses an additive $\sqrt{T \log^2 T}$ factor, achieving better performance when $M_{\mathcal{P}}(x_{1:T})$ is large. Here, we assume that $R_{\mathcal{B}}^{\mathbf{tr}}(T, \mathcal{F})$ is a concave function, which can be extended to any sublinear functions by standard results.

Next, we extend the above algorithm for $\mathcal{P}$ with the $\varepsilon$-ball metric, which is specific for our regression setting. Suppose that the prediction is outside the $\varepsilon$-ball at times $t_1, \ldots t_c \in [T]$, i.e., $\mathrm{d}(\mathcal{P}(x_{1:t-1})_t, x_t) \geq \varepsilon$ for $t \in \{t_1, \ldots t_c\}$, then we run a separate transductive online learner $\mathcal{B}$ for each duration $t_j, t_j + 1 \ldots, t_{j+1}$ for $j \in [c]$, i.e., we start a new instance whenever the prediction is outside the $\varepsilon$-ball. Since the prediction is always inside the $\varepsilon$-ball between $t_j$ and $t_{j+1}$, then if the function class is $L$-Lipschitz, our error bound has an additional $\varepsilon L T$ factor. That is, the minimax expected regret is upper bounded by $M_{\mathcal{P}}(\varepsilon, x_{1:T})R_{\mathcal{B}}^{\mathbf{tr}}(T, \mathcal{F}) + \varepsilon L_{\mathbf{los}} L_{\mathbf{hyp}} \cdot T$. We note that this algorithm can also be improved by the equi-distant partition of the time interval and MWA, as discussed earlier which achieves better performance when $M_{\mathcal{P}}(\varepsilon, x_{1:T})$ is large.

The above online learners take $\varepsilon$ as an input, so it is desirable to implement with an $\varepsilon$ that gives the optimal minimax expected regret. Then, if we know the explicit formula of the measure of predictability $M_{\mathcal{P}}(\varepsilon, x_{1:T})$, we can first compute the optimal choice of $\varepsilon$ and then implement the algorithms. In contrast, when $M_{\mathcal{P}}(\varepsilon, x_{1:T})$ have a complicated structure that makes it impossible to identify this $\varepsilon$, we can "guess" the optimal $\varepsilon$ geometrically in the range of $(0, \mathrm{poly}(T))$. That is, we

run MWA using experts with all $\varepsilon \in \{2^i, 2^i < \text{poly}(T), i \in \mathbb{Z}\}$ as inputs. This achieves the same asymptotic bound as the optimal $\varepsilon$ when $M_{\mathcal{P}}(\varepsilon, x_{1:T})$ has linear or polynomial dependency on $\varepsilon$, ensuring the effectiveness of our algorithm in real-world applications. Next, we present the explicit minimax expected regret under both metrics, representing the minimax expected regret as a function of the mistake-bounds of the Predictor.

### 4.1 EXPLICIT BOUNDS ON THE MINIMAX EXPECTED REGRET

In this section, we provide sufficient conditions on the mistake-bound of the Predictor to enable faster rates compared to online learning in the worst-case scenario. As a result, we identify function classes that are online learnable with predictions but not online learnable otherwise.

**Minimax regret under zero-one metric**  We first consider the minimax expected regret under zero-one metric. Assuming that the Predictor has a rate of $\tilde{\mathcal{O}}(T^p)$ where $p < 1$, the following theorem derives the upper bound on the minimax expected regret. We defer the proof to Appendix D.3.

**Theorem 4.2** (Zero-one metric). *Let $x_{1:T}$ be a sequence of examples, let $y_{1:T}$ be a sequence of labels, and let $\ell$ be the $\ell_1$-loss. Suppose that there is a Predictor that satisfies $M_{\mathcal{P}}(x_{1:T}) = \tilde{\mathcal{O}}(T^p)$, then for any function class $\mathcal{F} \subset [0,1]^{\mathcal{X}}$, there is an online learner $\mathcal{A}$ with minimax expected regret at most $\tilde{\mathcal{O}}(T^p) R_{\mathcal{B}}^{\mathbf{tr}}(T^{1-p}, \mathcal{F}) + \sqrt{T \log^2 T}$.*

We remark that a Predictor satisfying the mistake-bound conditions in Theorem 4.2 is possible if, for example, the sequence of examples $x_t \in \mathbb{R}^n$ are generated by a noise-free linear dynamical system (LDS) where system identification is possible in finite time. See Van Overschee & De Moor (2012); Green & Moore (1986) for further discussion for sufficient conditions under which system identification is possible.

As a Corollary, our next result shows that the minimax expected regret for the class of functions $\mathcal{F}^*$ on $[0,1]$ with bounded variation is roughly $T^{\frac{1+p}{2}}$, whose proof is deferred to Appendix D.4.

**Corollary 4.3** (Function class with bounded variation, zero-one metric). *Let $\mathcal{F}^*$ be a set of functions $f : [0,1] \to [0,1]$ with total variation of at most $V$, let $x_{1:T} \subset [0,1]$ be a sequence of examples, let $y_{1:T}$ be a sequence of labels, and let $\ell$ be the $\ell_1$-loss. Suppose that there is a Predictor that satisfies $M_{\mathcal{P}}(x_{1:T}) = \tilde{\mathcal{O}}(T^p)$, then there is an online learner $\mathcal{A}$ with minimax expected regret satisfying $R^{\mathbf{ol}}(T, \mathcal{F}^*) = \tilde{\mathcal{O}}\left(T^{\frac{1+p}{2}}\right)$. That is, $\mathcal{F}^*$ is online learnable with predictions if $p < 1$.*

We assume that there is a Predictor $\mathcal{P}$ that satisfies, for any sequence $x_{1:T} \subset \mathcal{X}$, its mistake-bound is sublinear in $T$ under the zero-one metric. Then the class of functions with bounded variation is online learnable. This implies a gap between online regression with predictions and online regression in the worst-case scenario, since $\mathcal{F}^*$ has an infinite sequential fat-shattering dimension, which characterizes online learnability.

**Minimax regret under $\varepsilon$-ball metric.**  Now, we compute the minimax expected regret under the $\varepsilon$-ball metric. In the following theorem, we assume that the rate of Predictor is $M_{\mathcal{P}}(\varepsilon, x_{1:T}) = \tilde{\mathcal{O}}\left(\frac{T^p}{\varepsilon^q}\right)$, and we will specify the conditions for $p$ and $q$ to guarantee learnability later. The proof is deferred to Appendix D.5.

**Theorem 4.4** ($\varepsilon$-ball metric). *Let $x_{1:T}$ be a sequence of examples, let $y_{1:T}$ be a sequence of labels, and let $\ell$ be the $\ell_1$-loss. Suppose that there is a Predictor that satisfies $M_{\mathcal{P}}(\varepsilon, x_{1:T}) = \tilde{\mathcal{O}}\left(\frac{T^p}{\varepsilon^q}\right)$, then for any $L_{\mathbf{hyp}}$-Lipschitz function class $\mathcal{F} \subset [0,1]^{\mathcal{X}}$, there is an online learner $\mathcal{A}$ with minimax expected regret at most*

$$\inf_{\varepsilon > 0} \left\{ \tilde{\mathcal{O}}\left(\frac{T^p}{\varepsilon^q}\right) R_{\mathcal{B}}^{\mathbf{tr}}\left(\varepsilon^q T^{1-p}, \mathcal{F}\right) + \varepsilon L_{\mathbf{hyp}} \cdot T + \sqrt{T \log^2 T} \right\}.$$

Like before, we remark that a Predictor satisfying the conditions in Theorem 4.4 can be constructed, if for example, the sequence of examples $x_t \in \mathbb{R}^n$ are generated by a noise-less dynamical system which need not be perfectly identifiable, but identifiable up to an error of $\varepsilon$ in finite time (Jansson & Wahlberg, 1998; Hazan et al., 2017).

The next statement shows the minimax expected regret for the class of functions $\mathcal{F}^*$ on $[0, 1]$ with bounded variation. The proof is deferred to Appendix D.6.

**Corollary 4.5** (Function class with bounded variation, $\varepsilon$-ball metric)**.** *Let $\mathcal{F}^*$ be a set of $L_{\mathbf{hyp}}$-Lipschitz $f : [0, 1] \to [0, 1]$ with total variation of at most $V$, let $x_{1:T} \subset [0, 1]$ be a sequence of examples, let $y_{1:T}$ be a sequence of labels, and let $\ell$ be the $\ell_1$-loss. Suppose that there is a Predictor that satisfies $M_{\mathcal{P}}(\varepsilon, x_{1:T}) = \tilde{\mathcal{O}}\left(\frac{T^p}{\varepsilon^q}\right)$, then there is an online learner $\mathcal{A}$ with minimax expected regret satisfying*

$$R^{\mathbf{ol}}(T, \mathcal{F}^*) = \tilde{\mathcal{O}}\left(L_{\mathbf{hyp}}^{\frac{q}{q+2}} \cdot T^{\frac{p+q+1}{q+2}}\right).$$

*That is, if the length of sequence is chosen as $T^c \operatorname{polylog}(T) \geq L_{\mathbf{hyp}}$ for some constant $c$, then $\mathcal{F}^*$ is online learnable with prediction if $p + cq < 1$.*

This result implies that the minimax expected regret of our algorithm scales with the quality of the Predictor. Now, we construct a class of functions with bounded variation, such that it is not online learnable in the worst case but online learnable given a good Predictor. Unfortunately, our algorithm requires Lipschitzness of the function classes, and the sequential fat-shattering dimension is equivalence to the fat-shattering dimension for Lipschitz classes up to negligible factors, so we do not achieve better rates for general Lipschitz classes.

The key observation here is that the rate of the transductive online learner for functions with bounded variation has no dependence on the Lipschitz factors (see Corollary C.9). Thus, if the error scale of the Predictor is sufficiently small, e.g., the extreme case when $M_{\mathcal{P}}(\varepsilon, x_{1:T}) = \mathcal{O}(1)$, we get rid of the Lipschitz dependence of the minimax expected regret of our algorithm. Then, we observe a gap between online learning in the worst case and online learning with prediction for classes with large Lipschitz constants. For instance, we consider the following class of ramp functions.

**Definition 4.6** (Class of ramp functions)**.** *We define the class of ramp functions to consist of all functions*

$$f_{a,b}(x) = \begin{cases} 0 & \text{if } 0 < x < a, \\ \frac{x-a}{b-a} & \text{if } a \leq x \leq b, \\ 1 & \text{if } b < x < 1, \end{cases}$$

*with $0 < a < b < 1$ and $b = a + \frac{1}{M}$.*

Note that by Theorem B.3 the minimax expected regret is roughly $\sqrt{L_{\mathbf{hyp}}T} = \sqrt{MT}$ in the online setting, then suppose that the length of sequence $T$ is chosen as $T = M$ by the adversary, we have no guarantee of the learnability of the function class in Definition 4.6 in the worst-case scenario. However, suppose that we have a Predictor satisfying $p + q < 1$, then by Corollary 4.5, the function class is learnable with prediction using our online learners.

## 5 CONCLUSIONS

In this paper, we study the problem of online regression in both the transductive and learning-augmented settings. In the transductive setting, we establish near-tight bounds on the minimax expected regret under the $\ell_1$-loss, showing that it is characterized by the fat-shattering dimension rather than the more restrictive sequential fat-shattering dimension. This separates transductive online learnability with online learnability for several critical function classes. In the online regression with predictions setting, we design algorithms whose regret adapts smoothly to the quality of the Predictor, interpolating between the worst-case and the transductive regime. We identify sufficient conditions on the Predictor that ensure the online learnability.

Our results provide a unified theoretical framework that separates these settings for online regression and opens several directions for future work, including empirical validation of our methods and further exploration of Predictors under general metrics. To complement our theoretical analysis, we included in the Appendix experiments on specific function classes. These illustrate that having prior information of the sequence of examples enables better empirical performance. While our small-scale experiments highlight the potential practical gains of our framework, a more systematic and large-scale empirical validation remains an important direction for future work. In addition, extending our framework to accommodate Predictors under more general metrics could yield deeper insights into the practical effectiveness of the learning-augmented online regression framework.

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

## A   Additional Related Work

In this section, we review additional related work.

**Algorithms with predictions.**   Machine learning models have achieved remarkable success across a wide range of application domains, often delivering predictions and decisions of impressive quality in practice. However, despite these empirical advances, such models rarely come with provably correct worst-case guarantees, and can result in embarrassingly inaccurate predictions when generalizing to previously unseen distributions (Szegedy et al., 2014). Learning-augmented algorithms (Mitzenmacher & Vassilvitskii, 2020) combine predictive models with principled algorithmic design to achieve provable worst-case guarantees while still benefiting from the strengths of data-driven approaches. The line of work most relevant to our setting is the direction of online algorithms with predictions, which achieve better performance than information-theoretic limits (Anand et al., 2021; 2022; Khodak et al., 2022; Antoniadis et al., 2023b). Specific applications include ski rental (Purohit et al., 2018; Gollapudi & Panigrahi, 2019; Anand et al., 2020; Wang et al., 2020; Wei & Zhang, 2020; Shin et al., 2023), scheduling, caching, and paging (Lattanzi et al., 2020; Lykouris & Vassilvitskii, 2021; Scully et al., 2022), covering and packing (Bamas et al., 2020; Im et al., 2021; Grigorescu et al., 2022; 2024), various geometric and graph objectives (Aamand et al., 2022; Almanza et al., 2021; Azar et al., 2022; Jiang et al., 2022; Antoniadis et al., 2023a). More recently, Eliás et al. (2024) proposed a model in which the Predictor can learn and adjust its predictions dynamically based on the data observed during execution. This approach differs from earlier work on learning-augmented online algorithms, where predictions are generated by machine learning models trained solely on historical data and remain fixed, lacking adaptability to the current input sequence. Eliás et al. (2024) examined several fundamental problems, such as caching and scheduling, demonstrating that carefully designed adaptive Predictors can yield stronger performance guarantees. We adopt a similar model established by (Raman & Tewari, 2024), where

we assume our algorithms have black-box access to an external mechanism that produces predictions about the examples, which evolve in response to the actual data encountered by the learning algorithm.

**Online classification.**    Motivated by applications such as spam filtering, image recognition, and language modeling, online classification has a rich history in statistical learning theory. Building on foundational concepts like the VC dimension (Vapnik & Chervonenkis, 1971) that characterize learnability in batch settings, (Littlestone, 1987) introduced the Littlestone dimension to precisely characterizes which binary hypothesis classes are online learnable in the realizable setting. This was subsequently extended to agnostic learning (Ben-David et al., 2009) and multiclass settings with finite and unbounded label spaces (Daniely et al., 2011; Hanneke et al., 2023a). More recently, (Hanneke et al., 2023b; 2024b) developed the theory of transductive online classification, providing performance guarantees for both binary and multiclass problems where the learner has access to the entire unlabeled input sequence before making predictions.

Unfortunately, the Littlestone dimension is often regarded as an impossibility result, as it rules out even simple classes such as threshold functions for online learning. This barrier is due to worst-case analyses where an adversary can select any sequence of labeled examples, potentially adapting future choices based on the previous learner choices. In practice, however, many data sequences are far from adversarial and often exhibit regularity or structure that worst-case models fail to capture. For example, when predicting short-term stock price movements, prices often follow temporal trends and correlations that predictive models can exploit. Motivated by beyond-worst-case analyses of such scenarios, Raman & Tewari (2024) explored online classification with predictive advice, showing that learning-augmented algorithms can achieve improved performance given predictions about the examples that need to be classified, complementing transductive frameworks.

**Online regression.**    While online regression extends online classification to sequential prediction with continuous outcomes, analyzing the complexity of the corresponding real-valued function classes often requires different tools than in classification. The fat-shattering dimension (Alon et al., 1997) generalizes the VC dimension to continuous-valued functions and plays a key role in bounding sample complexity and learnability. Data-dependent capacity measures such as Rademacher complexity (Bartlett & Mendelson, 2002) have been adapted to online settings through sequential Rademacher complexity and sequential fat-shattering dimension (Rakhlin et al., 2010; 2015a), capturing the difficulty of learning under adversarial data streams. More recent advances use generic chaining and majorizing measures to tightly control worst-case sequential Rademacher complexity (Block et al., 2021), establishing sharp uniform convergence and bounds that are robust to adversarial data. However, these results share a common limitation: they provide worst-case guarantees that often fail to capture improved performance on "easy" or structured data sequences.

**Smoothed Online Learning.**    In addition to auxiliary predictions, *smoothed analysis* is another framework for studying beyond-worst-case guarantees (Spielman & Teng, 2004; 2009). In the context of online regression, smoothed analysis involves placing some distributional assumptions on the data generation process. In particular, in each round, a smoothed adversary must choose and sample from a distribution belonging to a sufficiently anti-concentrated family of distributions. This allows one to go beyond the worst-case results in the fully adversarial model, where the adversary can pick any sequence of examples.

In the past couple years, there have been flurry papers studying online learnability under a smoothed adversary (Rakhlin et al., 2011; Haghtalab, 2018; Haghtalab et al., 2020; Block et al., 2022; Haghtalab et al., 2022; Blanchard, 2025; Blanchard & Kpotufe, 2025). We review the most relevant ones here. In the context of binary classification, Haghtalab (2018) and Haghtalab et al. (2020) show that this restriction on the adversary is enough for the VC dimension of a binary hypothesis class to be sufficient for online learnability. Block et al. (2022) and Blanchard (2025) extend these results to online regression and prove an analogous result – under a smoothed adversary, the fat-shattering dimension is sufficient for online learnability. In both cases, these results show that by placing some distributional assumptions on the input, online learning becomes as easy as batch learning. In this paper, we show a similar phenomena without needing any distributional assumptions on the examples, but given predictions about the future examples we will need to make predictions about.

# B  COMPLEXITY MEASURES

This section introduces several measures that are used to characterize the complexity of various function classes. We begin with the notions in the offline setting.

**Offline setting.** Given a function class $\mathcal{F} \subset \mathcal{Y}^{\mathcal{X}}$, we introduce the classical notion of the covering number, which defines the "effective" size of the function class on $\mathcal{X}$. A set $\mathcal{V} \subset \mathbb{R}^T$ is an $\alpha$-cover on a sequence $x = x_{1:T}$ with respect to the $\ell_p$ norm if for each $f \in \mathcal{F}$, there exist an item $v_{1:T} \in \mathcal{V}$ such that

$$\forall t \in [T], \quad \left( \frac{1}{T} \sum_{t=1}^{T} |f(x_t) - v_t|^p \right)^{1/p} \leq \alpha.$$

Similarly, a set $\mathcal{V} \subset \mathbb{R}^T$ is a $\alpha$-cover on a sequence $x = x_{1:T}$ with respect to the $\ell_\infty$ norm if for each $f \in \mathcal{F}$, there exist an item $v_{1:T} \in \mathcal{V}$ such that

$$\forall t \in [T], \quad |f(x_t) - v_t| \leq \alpha.$$

Then the covering number $\mathcal{N}_p(T, \mathcal{F}, x, \alpha)$ of $\mathcal{F}$ on $x$ is defined as the minimum size of the $\alpha$-cover with respect to the $\ell_p$ norm. It is known that $\mathcal{N}_p(T, \mathcal{F}, x, \alpha) \leq \mathcal{N}_q(T, \mathcal{F}, x, \alpha)$ for $1 \leq p \leq q \leq \infty$. In addition, we define the covering number $\mathcal{N}_p(T, \mathcal{F}, \alpha)$ of $\mathcal{F}$ on the example space $\mathcal{X}$ as the supremum of all choices of sequence $x$: $\mathcal{N}_p(T, \mathcal{F}, \alpha) = \sup_{x \subset \mathcal{X}} \mathcal{N}_p(T, \mathcal{F}, x, \alpha)$.

Next, we introduce the fat-shattering dimension, which is a scale-sensitive version of the Vapnik-Chervonenkis (VC) dimension (Vapnik & Chervonenkis, 1971), and is used when the range of the function class is a real interval, e.g. $[0, 1]$. We say that a sequence $x = x_{1:T}$ is $\alpha$-shattered by $\mathcal{F}$ if there exists a sequence of real numbers $y = y_{1:T}$ such that for each binary string $\sigma \in \{-1, 1\}^T$, there is a function $f \in \mathcal{F}$ that satisfies

$$\forall t \in [T], \quad \sigma_t \cdot (f(x_t) - y_t) \geq \alpha/2.$$

Here, the sequence $y$ is called the witness of shattering. Then, the fat-shattering dimension $\mathrm{fat}_\alpha(\mathcal{F})$ is defined as the largest $T$ such that $\mathcal{F}$ $\alpha$-shatters a sequence $x \subset \mathcal{X}$ of length $T$. In addition, $\mathrm{fat}_\alpha(\mathcal{F}) = \infty$ if for every finite $T$ there is a sequence of length $T$ that is $\alpha$-shattered by $\mathcal{F}$. The next statement upper bounds the $\alpha$-covering number by the fat-shattering dimension.

**Theorem B.1** (See Theorem 12.8 in (Anthony & Bartlett, 1999)). *For any function class $\mathcal{F} \subset [0, 1]^{\mathcal{X}}$ and $\alpha > 0$, we have*

$$\log \mathcal{N}_\infty(T, \mathcal{F}, \alpha) \leq \mathrm{fat}_{\alpha/4}(\mathcal{F}) \cdot c \log^2 \frac{T}{\alpha},$$

*where $c$ is some universal constant.*

We then introduce the Rademacher complexity of a function class, which is closely related to the fat-shattering dimension. Let $\sigma_1, \dots, \sigma_T$ be independent Rademacher random variables. We define the Rademacher complexity of a function class $\mathcal{F} \subset [0, 1]^{\mathcal{X}}$ on an example sequence $x_{1:T}$ as

$$\mathcal{R}(T, \mathcal{F}, x) = \mathbb{E} \left[ \sup_{f \in \mathcal{F}} \frac{1}{T} \sum_{t=1}^{T} \sigma_t f(x_t) \right].$$

Then, we define the Rademacher complexity as $\mathcal{R}(T, \mathcal{F}) = \sup_{x_{1:T} \subset \mathcal{X}} \mathcal{R}(T, \mathcal{F}, x)$. We state the entropy bound on the Rademacher complexity in the following statement. Together with Theorem B.1, it gives an upper bound on Rademacher complexity by the fat-shattering dimension, which is used to characterize the learnability of tranductive online regression in later sections.

**Theorem B.2** (See Theorem 12.4 in (Rakhlin & Sridharan, 2014)). *For any sequence $x_{1:T}$ and function class $\mathcal{F} \subset [0, 1]^{\mathcal{X}}$, we have*

$$\mathcal{R}(T, \mathcal{F}, x) \leq \inf_{\alpha \geq 0} \left( 4\alpha + \frac{12}{\sqrt{T}} \int_\alpha^1 \sqrt{\log \mathcal{N}_2(T, \mathcal{F}, x, \beta)} \, d\beta \right).$$

**Online setting.** Unlike the offline setting with i.i.d. examples, the (adversarially) online learning setting has a sequential dependence, which is not captured by the offline definitions. We begin with the notion of the Littlestone tree (Littlestone, 1987; Rakhlin & Sridharan, 2014; Rakhlin et al., 2015a; Raman & Tewari, 2024), which encrypts the sequentially dependence of the examples. A $\mathcal{X}$-valued Littlestone tree $x$ of depth $T$ is a complete binary tree of depth $T$, where the nodes are labeled by examples $x \in \mathcal{X}$ and the outgoing edges to the left and right of the nodes are labeled by $-1$ and $1$ respectively. Given a binary string $\sigma = (\sigma_1, \ldots, \sigma_T) \in \{-1, 1\}^T$ of depth $T$, we define a path $x(\sigma)$ induced by $\sigma$ to be $\{(x_i, \sigma_i)\}_{i=1}^T$, where $x_i$ is the example labeling the node following the prefix edges $(\sigma_1, \ldots, \sigma_{i-1})$ down the tree. For simplicity, we use $x_t(\sigma)$ to denote the example at the $t$-th entry of a path $x(\sigma)$, but we remark that $x_t(\sigma)$ depends only on the prefix path $(\sigma_1, \ldots, \sigma_{i-1})$.

Following this notion, we introduce the sequential version of the fat-shattering dimension. We say that a $\mathcal{X}$-valued Littlestone tree of depth $T$ is $\alpha$-shattered by a function class $\mathcal{F} \subset \mathcal{Y}^{\mathcal{X}}$ if there exists a $\mathcal{Y}$-valued tree $y$ of depth $T$ such that

$$\forall \sigma \in \{-1, 1\}^T, \ \exists f \in \mathcal{F}, \ \text{s.t.} \ \forall t \in [T], \ \sigma_t(f(x_t(\sigma)) - y_t(\sigma)) \geq \alpha/2.$$

Here, the tree $y$ is called the witness of shattering. Similar as the offline definition, the sequential fat-shattering dimension $\text{fat}_\alpha^{\textbf{seq}}(\mathcal{F})$ is defined as the largest $T$ such that $\mathcal{F}$ $\alpha$-shatters a $\mathcal{X}$-valued tree of depth $T$. In addition, $\text{fat}_\alpha^{\textbf{seq}}(\mathcal{F}) = \infty$ if for every finite $T$, there is a $\mathcal{X}$-valued tree of depth $T$ that is $\alpha$-shattered by $\mathcal{F}$. The following statement shows that the minimax expected regret of online learning is controlled by the sequential fat-shattering dimension.

**Theorem B.3** (Online learning, see Proposition 9 in (Rakhlin et al., 2015a)). *For any function class $\mathcal{F} \subset [0, 1]^{\mathcal{X}}$ and $L_{\textbf{los}}$-Lipschitz and convex loss function $\ell$, the minimax expected regret of online learning satisfies*

$$\inf_{\mathcal{A}} R_{\mathcal{A}}^{\textbf{ol}}(T, \mathcal{F}) \leq 2L_{\textbf{los}}T \cdot \inf_{\alpha \geq 0} \left( 4\alpha + \frac{12}{\sqrt{T}} \int_\alpha^1 \sqrt{\text{fat}_{\beta/4}^{\textbf{seq}}(\mathcal{F}) \log \frac{2eT}{\beta}} \, \mathrm{d}\beta \right),$$

*where the infimum is taken on all online learner $\mathcal{A}$.*

## C  MINIMAX REGRET OF TRANSDUCTIVE ONLINE REGRESSION

In this section, we provide additional discussions and missing proofs in Section 3.

### C.1  TRANSDUCTIVE ONLINE LEARNER

In this section, we present a concrete learning algorithm based on the multiplicative weights algorithm (MWA), which randomly samples the advice of $K$ experts in an online manner. Since we know the exact input sequence of examples $x_{1:T}$, we define the $K$ experts in MWA by the minimal $\alpha$-cover on $x_{1:T}$ with respect to the $\ell_\infty$ norm. We remark that this construction does not match our optimal upper bound on the minimax expected regret, but it provides intuitions on how we use the knowledge of the sequence of examples: When we know $x_{1:T}$, we can build a net with better coverage to apply canonical algorithms in online learning. Our transductive online learner is presented in Algorithm 1. Next, we state the formal theorem for MWA.

**Theorem C.1** (See e.g. Section 4 in (Cesa-Bianchi & Lugosi, 2006) and Theorem 2.1 in (Arora et al., 2012)). *Given $K$ experts such that for each expert $k \in K$, the loss in each round $t$ is $\ell(k_t, y_t)$, suppose that the loss ranges from $[0, 1]$, then there is a multiplicative weights algorithm $\mathcal{Q}$ with minimax expected regret $\eta = \sqrt{\frac{8 \log K}{T}}$ that satisfies*

$$\mathbb{E}\left[ \sum_{t=1}^T \ell(\mathcal{Q}(x_t), y_t) \right] \leq \inf_{k \in [K]} \left( \sum_{t=1}^T \ell(k_t, y_t) \right) + \sqrt{\frac{T \log K}{2}}.$$

The next theorem upper bounds the minimax expected regret of Algorithm 1.

**Theorem C.2.** *For any function class $\mathcal{F} \subset [0, 1]^{\mathcal{X}}$, $L_{\textbf{los}}$-Lipschitz loss function $\ell$, set of examples $\{x_1, \ldots, x_T\}$, and sequence of labels $y_1, \ldots, y_T$, the minimax expected regret of Algorithm 1 is*

---

**Algorithm 1** Transductive Online Learner

---

1: **Input:** Function class $\mathcal{F}$, time interval $[T]$, covering parameter $\alpha$, sequence of examples $x_{1:T}$ revealed at initialization, sequence of labels $y_{1:T}$ revealed sequentially
2: **Output:** Predictions to $y_{1:T}$
3: Let $\mathcal{V} = \{v^1 \ldots, v^K\}$ be the $\alpha$-$\ell_\infty$-cover on $x_{1:T}$ with minimum size
4: Define the expert $k$ to be $v^k$ for each $k \in [K]$
5: **for** $t \in [T]$ **do**
6:     **Return:** Prediction from MWA $\mathcal{Q}$ with the $K$ experts {See Theorem C.1}
7:     Reveal the actual label $y_t$ from the adversary and input into $\mathcal{Q}$
8: **end for**

---

*bounded by*

$$\inf_{\alpha > 0} \left( \alpha L_{\mathbf{los}} T + \sqrt{\frac{T \cdot \mathrm{fat}_{\alpha/4}(\mathcal{F}) \cdot c \log^2 \frac{T}{\alpha}}{2}} \right),$$

*where $c$ is a universal constant.*

*Proof.* Let $\mathcal{B}$ be the learner in Algorithm 1. From the guarantee of MWA (see Theorem C.1), we have

$$\mathbb{E}\left[ \sum_{t=1}^T \ell(\mathcal{B}(x_t), y_t) \right] \leq \inf_{k \in [K]} \left( \sum_{t=1}^T \ell(k_t, y_t) \right) + \sqrt{\frac{T \log K}{2}}$$

$$= \inf_{v^k \in \mathcal{V}} \left( \sum_{t=1}^T \ell(v_t^k, y_t) \right) + \sqrt{\frac{T \log K}{2}}.$$

Recall that $\mathcal{V}$ is a $\alpha$-cover of $\mathcal{F}$ on $x_{1:T}$, then for each $f \in \mathcal{F}$, there is a $v^k \in \mathcal{V}$ such that $|v_t^k - f(x_t)| < \alpha$ for each $t \in [T]$. Since the loss function is $L_{\mathbf{los}}$-Lipschitz, we have

$$\ell(v_t^k, y_t) \leq \ell(f(x_t), y_t) + \alpha L_{\mathbf{los}}.$$

Summing over each $t \in [T]$ gives us

$$\sum_{t=1}^T \ell(v_t^k, y_t) \leq \sum_{t=1}^T \ell(f(x_t), y_t) + \alpha L_{\mathbf{los}} T.$$

Therefore, for each $f \in \mathcal{F}$, we have

$$\inf_{v^k \in \mathcal{V}} \left( \sum_{t=1}^T \ell(v_t^k, y_t) \right) \leq \sum_{t=1}^T \ell(f(x_t), y_t) + \alpha L_{\mathbf{los}} T.$$

Taking the infimum across $h$ on the RHS, we have

$$\inf_{v^k \in \mathcal{V}} \left( \sum_{t=1}^T \ell(v_t^k, y_t) \right) \leq \inf_{f \in \mathcal{F}} \left( \sum_{t=1}^T \ell(f(x_t), y_t) \right) + \alpha L_{\mathbf{los}} T.$$

Then the expected loss satisfies

$$\mathbb{E}\left[ \sum_{t=1}^T \ell(\mathcal{B}(x_t), y_t) \right] \leq \inf_{f \in \mathcal{F}} \left( \sum_{t=1}^T \ell(f(x_t), y_t) \right) + \alpha L_{\mathbf{los}} T + \sqrt{\frac{T \log K}{2}}.$$

Therefore, by definition the minimax expected regret of $\mathcal{B}$ is at most

$$\alpha L_{\mathbf{los}} T + \sqrt{\frac{T \log K}{2}}.$$

Last, we note that $K \leq \mathcal{N}_\infty(T, \mathcal{F}, \alpha)$ since we define the experts by a minimum $\alpha$-cover $\mathcal{F}'$ on $x_{1:T}$ (see the definitions in Appendix B). By Theorem B.1, we have

$$\log \mathcal{N}_\infty(T, \mathcal{F}, \alpha) \leq \mathrm{fat}_{\alpha/4}(\mathcal{F}) \cdot c \log^2 \frac{T}{\alpha}.$$

Therefore, the minimax expected regret is at most

$$\alpha L_{\mathbf{los}}T + \sqrt{\frac{T \cdot \mathrm{fat}_{\alpha/4}(\mathcal{F}) \cdot c \log^2 \frac{T}{\alpha}}{2}}.$$

Our result then follows from the fact that the above equation holds for an arbitrary $\alpha$. □

### C.2 Proof of the Upper Bound in Theorem 3.1

*Proof.* Let $\ell'(\widehat{y}_t, y_t)$ be a subgradient of the function $y \to \ell(\cdot, y_t)$ at $y_t$. Then, since the loss function is convex, we have

$$\inf_{\mathcal{A}} R_{\mathcal{A}}^{\mathbf{tr}}(T, \mathcal{F}) \leq \sup_{x_{1:T}} \inf_{q_1 \in \mathcal{Q}} \sup_{y_1 \in \mathcal{Y}} \mathop{\mathbb{E}}_{\mathcal{A}_1 \sim q_1} \cdots \inf_{q_T \in \mathcal{Q}} \sup_{y_T \in \mathcal{Y}} \mathop{\mathbb{E}}_{\mathcal{A}_T \sim q_T} \left[ \sup_{f \in \mathcal{F}} \sum_{t=1}^{T} \ell'(\mathcal{A}_t, y_t) \cdot (\mathcal{A}_t - f(x_t)) \right].$$

In addition, since the loss function satisfies the Lipschitz property, i.e., $|\ell'(\mathcal{A}_t, y_t)| < L_{\mathbf{los}}$, then we have

$$\inf_{\mathcal{A}} R_{\mathcal{A}}^{\mathbf{tr}}(T, \mathcal{F}) \leq \sup_{x_{1:T}} \inf_{q_1 \in \mathcal{Q}} \sup_{y_1 \in \mathcal{Y}} \mathop{\mathbb{E}}_{\mathcal{A}_1 \sim q_1} \sup_{s_1 \in [-L,L]} \cdots \inf_{q_T \in \mathcal{Q}} \sup_{y_T \in \mathcal{Y}} \mathop{\mathbb{E}}_{\mathcal{A}_T \sim q_T} \sup_{s_T \in [-L,L]} \left[ \sup_{f \in \mathcal{F}} \sum_{t=1}^{T} s_t \cdot (\mathcal{A}_t - f(x_t)) \right].$$

Here, we write $L_{\mathbf{los}}$ as $L$ for simplicity of notation. We then simplify the above upper bound as follows since $y_t$ does not appear in the objective function

$$\sup_{x_{1:T}} \inf_{q_1 \in \mathcal{Q}} \mathop{\mathbb{E}}_{\mathcal{A}_1 \sim q_1} \sup_{s_1 \in [-L,L]} \cdots \inf_{q_T \in \mathcal{Q}} \mathop{\mathbb{E}}_{\mathcal{A}_T \sim q_T} \sup_{s_T \in [-L,L]} \left[ \sup_{f \in \mathcal{F}} \sum_{t=1}^{T} s_t \cdot (\mathcal{A}_t - f(x_t)) \right].$$

Next, since the family of probability measures $\mathcal{Q}$ contains the point distribution, we can write the operator $\inf_{q_t \in \mathcal{Q}} \mathop{\mathbb{E}}_{\mathcal{A}_t \sim q_t}$ as $\inf_{\mathcal{A}_t \in [0,1]}$. Similarly, let $\mathcal{P}$ denote the family of all possible distributions on $[-L, L]$, we can write $\sup_{s_t \in [-L,L]}$ as $\sup_{p_t \in \mathcal{P}} \mathop{\mathbb{E}}_{s_t \sim p_t}$. Then, the upper bound is equivalent to

$$\sup_{x_{1:T}} \inf_{\mathcal{A}_1 \in [0,1]} \sup_{p_1 \in \mathcal{P}} \mathop{\mathbb{E}}_{s_1 \sim p_1} \cdots \inf_{\mathcal{A}_T \in [0,1]} \sup_{p_T \in \mathcal{P}} \mathop{\mathbb{E}}_{s_T \sim p_T} \left[ \sum_{t=1}^{T} s_t \cdot \mathcal{A}_t - \inf_{f \in \mathcal{F}} \sum_{t=1}^{T} s_t \cdot f(x_t) \right].$$

Notice that $\mathop{\mathbb{E}}_{s_t \sim p_t} \left[ \sum_{t=1}^{T} s_t \cdot \mathcal{A}_t - \inf_{f \in \mathcal{F}} \sum_{t=1}^{T} s_t \cdot f(x_t) \right]$ is concave in $p_T$ and convex in $\mathcal{A}_T$, then by the minimax theorem, we have

$$\inf_{\mathcal{A}_T \in [0,1]} \sup_{p_T \in \mathcal{P}} \mathop{\mathbb{E}}_{s_T \sim p_T} \left[ \sum_{t=1}^{T} s_t \cdot \mathcal{A}_t - \inf_{f \in \mathcal{F}} \sum_{t=1}^{T} s_t \cdot f(x_t) \right]$$

$$= \sup_{p_T \in \mathcal{P}} \inf_{\mathcal{A}_T \in [0,1]} \mathop{\mathbb{E}}_{s_T \sim p_T} \left[ \sum_{t=1}^{T} s_t \cdot \mathcal{A}_t - \inf_{f \in \mathcal{F}} \sum_{t=1}^{T} s_t \cdot f(x_t) \right]$$

$$= \sum_{t=1}^{T-1} s_t \cdot \mathcal{A}_t + \sup_{p_T \in \mathcal{P}} \mathop{\mathbb{E}}_{s_T \sim p_T} \left[ \inf_{\mathcal{A}_T \in [0,1]} \mathop{\mathbb{E}}_{s_T \sim p_T} s_T \cdot \mathcal{A}_T - \inf_{f \in \mathcal{F}} \sum_{t=1}^{T} s_t \cdot f(x_t) \right].$$

Similarly, $\mathop{\mathbb{E}}_{s_T \sim p_T} \left[ \inf_{\mathcal{A}_T \in [0,1]} \mathop{\mathbb{E}}_{s_T \sim p_T} s_t \cdot \mathcal{A}_t - \inf_{f \in \mathcal{F}} \sum_{t=1}^{T} s_t \cdot f(x_t) \right]$ is concave in $p_{T-1}$ and convex in $\mathcal{A}_{T-1}$, then again by the minimax theorem, we have

$$\sup_{p_{T-1} \in \mathcal{P}} \inf_{\mathcal{A}_{T-1} \in [0,1]} \mathop{\mathbb{E}}_{s_{T-1} \sim p_{T-1}} \left[ \sum_{t=1}^{T-1} s_t \cdot \mathcal{A}_t + \sup_{p_T \in \mathcal{P}} \mathop{\mathbb{E}}_{s_T \sim p_T} \left[ \inf_{\mathcal{A}_T \in [0,1]} \mathop{\mathbb{E}}_{s_T \sim p_T} s_t \cdot \mathcal{A}_t - \inf_{f \in \mathcal{F}} \sum_{t=1}^{T} s_t \cdot f(x_t) \right] \right]$$

$$= \inf_{\mathcal{A}_{T-1} \in [0,1]} \sup_{p_{T-1} \in \mathcal{P}} \mathop{\mathbb{E}}_{s_{T-1} \sim p_{T-1}} \left[ \sum_{t=1}^{T-1} s_t \cdot \mathcal{A}_t + \sup_{p_T \in \mathcal{P}} \mathop{\mathbb{E}}_{s_T \sim p_T} \left[ \inf_{\mathcal{A}_T \in [0,1]} \mathop{\mathbb{E}}_{s_T \sim p_T} s_t \cdot \mathcal{A}_t - \inf_{f \in \mathcal{F}} \sum_{t=1}^{T} s_t \cdot f(x_t) \right] \right]$$

$$= \sum_{t=1}^{T-2} s_t \cdot \mathcal{A}_t + \sup_{p_{T-1} \in \mathcal{P}} \mathop{\mathbb{E}}_{s_{T-1} \sim p_{T-1}} \sup_{p_T \in \mathcal{P}} \mathop{\mathbb{E}}_{s_T \sim p_T} \left[ \sum_{t=T-1}^{T} \inf_{\mathcal{A}_t \in [0,1]} \mathop{\mathbb{E}}_{s_t \sim p_t} s_t \cdot \mathcal{A}_t - \inf_{f \in \mathcal{F}} \sum_{t=1}^{T} s_t \cdot f(x_t) \right].$$

Proceeding with this transformation, we have that the minimax expected regret is upper bounded by

$$\sup_{x_{1:T}} \sup_{p_1 \in \mathcal{P}} \mathop{\mathbb{E}}_{s_1 \sim p_1} \cdots \sup_{p_T \in \mathcal{P}} \mathop{\mathbb{E}}_{s_T \sim p_T} \left[ \sum_{t=1}^{T} \inf_{\mathcal{A}_t \in [0,1]} \mathop{\mathbb{E}}_{s_t \sim p_t} s_t \cdot \mathcal{A}_t - \inf_{f \in \mathcal{F}} \sum_{t=1}^{T} s_t \cdot f(x_t) \right].$$

Replacing $\mathcal{A}_t$ by a potential suboptimal choice $f(x_t)$, we obtain an upper bound

$$\sup_{x_{1:T}} \sup_{p_1 \in \mathcal{P}} \mathop{\mathbb{E}}_{s_1 \sim p_1} \cdots \sup_{p_T \in \mathcal{P}} \mathop{\mathbb{E}}_{s_T \sim p_T} \left[ \sup_{f \in \mathcal{F}} \left[ \sum_{t=1}^{T} \left( \mathop{\mathbb{E}}_{s_t \sim p_t} s_t - s_t \right) \cdot f(x_t) \right] \right]$$

$$= \sup_{x_{1:T}} \sup_{p_1 \in \mathcal{P}} \mathop{\mathbb{E}}_{s_1, s_1' \sim p_1} \cdots \sup_{p_T \in \mathcal{P}} \mathop{\mathbb{E}}_{s_T, s_T' \sim p_T} \left[ \sup_{f \in \mathcal{F}} \left[ \sum_{t=1}^{T} (s_t' - s_t) \cdot f(x_t) \right] \right].$$

Since the objective function in the expectation is symmetric with respect to $s_t'$ and $s_t$, it equals to

$$\sup_{x_{1:T}} \sup_{p_1 \in \mathcal{P}} \mathop{\mathbb{E}}_{s_1, s_1' \sim p_1} \mathop{\mathbb{E}}_{\sigma_1} \cdots \sup_{p_T \in \mathcal{P}} \mathop{\mathbb{E}}_{s_T, s_T' \sim p_T} \mathop{\mathbb{E}}_{\sigma_T} \left[ \sup_{f \in \mathcal{F}} \left[ \sum_{t=1}^{T} \sigma_t (s_t' - s_t) \cdot f(x_t) \right] \right],$$

where $\sigma_t$ are Rademacher variables. Since $s_t \in [-L, L]$, we obtain an upper bound

$$\sup_{x_{1:T}} \sup_{s_1 \in [-2L, 2L]} \mathop{\mathbb{E}}_{\sigma_1} \cdots \sup_{s_T \in [-2L, 2L]} \mathop{\mathbb{E}}_{\sigma_T} \left[ \sup_{f \in \mathcal{F}} \left[ \sum_{t=1}^{T} \sigma_t s_t \cdot f(x_t) \right] \right].$$

Note that for each $t \in [T]$, the objective is convex in $s_t$, and so the supremum is achieved at the endpoints, therefore, we have an upper bound

$$\sup_{x_{1:T}} \sup_{s_1 \in \{-2L, 2L\}} \mathop{\mathbb{E}}_{\sigma_1} \cdots \sup_{s_T \in \{-2L, 2L\}} \mathop{\mathbb{E}}_{\sigma_T} \left[ \sup_{f \in \mathcal{F}} \left[ \sum_{t=1}^{T} \sigma_t s_t \cdot f(x_t) \right] \right]$$

$$= 2L \cdot \sup_{x_{1:T}} \sup_{s_1 \in \{-1, 1\}} \mathop{\mathbb{E}}_{\sigma_1} \cdots \sup_{s_T \in \{-1, 1\}} \mathop{\mathbb{E}}_{\sigma_T} \left[ \sup_{f \in \mathcal{F}} \left[ \sum_{t=1}^{T} \sigma_t s_t \cdot f(x_t) \right] \right].$$

Now, for an arbitrary function $g : \{\pm 1\} \to \mathbb{R}$, we have that

$$\sup_{s_T \in \{-1, 1\}} \mathop{\mathbb{E}}_{\sigma} [g(s\sigma)] = \sup_{s_T \in \{-1, 1\}} \frac{1}{2} g(s) + g(-s) = \mathop{\mathbb{E}}_{\sigma} [g(\sigma)].$$

Therefore, the above quantity equals to

$$2L \cdot \sup_{x_{1:T}} \mathop{\mathbb{E}}_{\sigma} \left[ \sup_{f \in \mathcal{F}} \left[ \sum_{t=1}^{T} \sigma_t f(x_t) \right] \right] = 2LT \cdot \mathcal{R}(T, \mathcal{F}),$$

which upper bounds the minimax expected regret by the Rademacher complexity. Combining Theorem B.1 and Theorem B.2, we have the following entropy bound on the Rademacher complexity, which is associated with the fat-shattering dimension.

$$\mathcal{R}(T, \mathcal{F}) \leq \inf_{\alpha \geq 0} \left( 4\alpha + \frac{12}{\sqrt{T}} \int_\alpha^1 \sqrt{\mathrm{fat}_{\beta/4}(\mathcal{F}) \cdot c \log^2 \frac{T}{\beta}} \, d\beta \right),$$

which gives our final result. $\qquad \square$

## C.3    Proof of the Lower Bound in Theorem 3.1

*Proof.* Our proof is inspired by the hard instance for transductive online binary classification (Hanneke et al., 2023b), where they construct the sequence of example by $k$ copies of sequence

$x_1^*, \ldots, x_d^*$ that is VC-shattered by the function class and then apply the anti-concentration property of Rademacher variables.

We assume that the label space is $[-1, 1]$ for simplicity of computation, which can be obtained by linear transformation. First, we consider the case when $\mathrm{fat}_\alpha(\mathcal{F}) = d < T$, and we assume $T = kd$, where $k$ is an integer. Let $\{x_1, \ldots, x_d\}$ be a sequence $\alpha$-shattered by $\mathcal{F}$. We define the input sequence of examples to be

$$x_1^1, \ldots, x_1^k, x_2^1, \ldots, x_2^k, \ldots x_d^1, \ldots, x_d^k,$$

where $x_i^1 = \cdots = x_i^k = x_i$ for each $i \in [d]$. We define the sequence of labels by generating i.i.d. random Rademacher variables, i.e., $y_t \in \{-1, 1\}$. Fix an arbitrary transductive learning algorithm $\mathcal{A}$, by the probabilistic method, it suffices to lower bound

$$\mathbb{E}_{\mathcal{A}, y \sim \{-1,1\}^T} \left[ \sum_{t=1}^T |\mathcal{A}(x_t) - y_t| - \min_{f \in \mathcal{F}} \sum_{t=1}^T |f(x_t) - y_t| \right].$$

First, note that we generate the random labels $y_t$ independently, and so

$$\mathbb{E}_{\mathcal{A}, y \sim \{-1,1\}^T} \left[ \sum_{t=1}^T |\mathcal{A}(x_t) - y_t| \right] = T.$$

Next, since we have $|a - y_t| = 1 - a y_t$ for any $a \in [-1, 1]$ and $y_t \in [-1, 1]$, we have

$$\mathbb{E}_{\mathcal{A}, y \sim \{-1,1\}^T} \left[ \min_{f \in \mathcal{F}} \sum_{t=1}^T |f(x_t) - y_t| \right] = T - \mathbb{E}_{y \sim \{-1,1\}^T} \left[ \max_{f \in \mathcal{F}} \sum_{t=1}^T f(x_t) y_t \right].$$

Therefore, we have

$$\mathbb{E}_{\mathcal{A}, y \sim \{-1,1\}^T} \left[ |\mathcal{A}(x) - y_t| - \min_{f \in \mathcal{F}} \sum_{t=1}^T |f(x_t) - y_t| \right] \geq \mathbb{E}_{y \sim \{-1,1\}^T} \left[ \max_{f \in \mathcal{F}} \sum_{t=1}^T f(x_t) y_t \right].$$

Let $\{s_1, \ldots, s_d\}$ be the witness of $\alpha$-shattering for set $\{x_1, \ldots, x_d\}$. Since $\mathbb{E}_{y \sim \{-1,1\}^T} \left[ \sum_{t=1}^T y_t s_{\lceil \frac{t}{k} \rceil} \right] = 0$, the above quantity is equal to

$$\mathbb{E}_{y \sim \{-1,1\}^T} \left[ \max_{f \in \mathcal{F}} \sum_{t=1}^T y_t (f(x_t) - s_{\lceil \frac{t}{k} \rceil}) \right] = \mathbb{E}_{y \sim \{-1,1\}^T} \left[ \max_{f \in \mathcal{F}} \sum_{i=1}^d \sum_{j=1}^k y_i^j (f(x_i^j) - s_i) \right].$$

Let $\sigma_i := \mathrm{sign}(\sum_{j=1}^k y_i^j)$, which is the majority vote of the signs $y_i^j$ in block $i$. Then, the above quantity is equal to

$$\mathbb{E}_{y \sim \{-1,1\}^T} \left[ \max_{f \in \mathcal{F}} \sum_{i=1}^d \left| \sum_{j=1}^k y_i^j \right| \sigma_i (f(x_i) - s_i) \right].$$

Due to the definition of $\alpha$-shattering, there exists a function $\bar{f} \in \mathcal{F}$ that satisfies $\sigma_i(\bar{f}(x_i) - s_i) \geq \alpha/2$ for each $i \in [d]$, then we have

$$\mathbb{E}_{y \sim \{-1,1\}^T} \left[ \max_{f \in \mathcal{F}} \sum_{i=1}^d \left| \sum_{j=1}^k y_i^j \right| \sigma_i (f(x_i) - s_i) \right] \geq \mathbb{E}_{y \sim \{-1,1\}^T} \left[ \sum_{i=1}^d \left| \sum_{j=1}^k y_i^j \right| \sigma_i (\bar{f}(x_i) - s_i) \right]$$

$$\geq \frac{\alpha}{2} \cdot \mathbb{E}_{y \sim \{-1,1\}^T} \left[ \sum_{i=1}^d \left| \sum_{j=1}^k y_i^j \right| \right]$$

$$= \frac{\alpha d}{2} \cdot \mathbb{E}_{y \sim \{-1,1\}^k} \left[ \left| \sum_{j=1}^k y_i^j \right| \right].$$

Then, by Khintchine's inequality, we have

$$\frac{\alpha d}{2} \cdot \mathbb{E}_{y \sim \{-1,1\}^k} \left[ \left| \sum_{j=1}^k y_i^j \right| \right] \geq \frac{\alpha d}{2} \cdot \sqrt{\frac{k}{2}} = \frac{\alpha d}{2} \cdot \sqrt{\frac{T}{2d}} = \alpha \cdot \sqrt{\frac{T \cdot \mathrm{fat}_\alpha(\mathcal{F})}{8}}.$$

Recall that we assume $T = kd$, now, for a general $T$, we take $T' = kd > T/2$ and apply the same analysis as above. Thus, we have

$$\mathcal{R}(T, \mathcal{F}) \geq \sup_{\alpha : \mathrm{fat}_\alpha(\mathcal{F}) < T} \frac{\alpha}{4} \cdot \sqrt{T \cdot \mathrm{fat}_\alpha(\mathcal{F})}$$

Last, for $\mathrm{fat}_\alpha(\mathcal{F}) \geq T$, we take the sequence of examples to be the set $\{x_1, \ldots, x_T\}$ $\alpha$-shattered by $\mathcal{F}$ with witness $\{s_1, \ldots, s_T\}$. Then, by the definition of $\alpha$-shattering, we have

$$\mathbb{E}_{y \sim \{-1,1\}^T} \left[ \sum_{t=1}^T y_t(f(x_t) - s_t) \right] \geq \frac{\alpha T}{2}.$$

Therefore, we have

$$R^{\mathbf{tr}}(T, \mathcal{F}) \geq \sup_\alpha \left( \frac{\alpha}{4} \cdot \sqrt{T \cdot \min\{\mathrm{fat}_\alpha(\mathcal{F}), T\}} \right).$$

$\square$

## C.4 APPLICATIONS TO FUNCTION CLASSES

In this section, we present additional explicit minimax expected regret for transductive online regression for Lipschitz functions, $k$-fold aggregations, and functions with bounded variation.

**Lipschitz function.** We consider the class of $L_{\mathbf{hyp}}$-Lipschitz functions. The next statement upper bounds the fat-shattering dimension for such classes.

**Theorem C.3.** *(See corollary 1 in (Gottlieb et al., 2014)) Let $\mathcal{X}$ be a metric space with diameter $\mathrm{diam}(\mathcal{X})$ and doubling dimension $\mathrm{ddim}(\mathcal{X})$. For any function class $\mathcal{F} \subset [0,1]^{\mathcal{X}}$ of $L_{\mathbf{hyp}}$-Lipschitz function and parameter $\alpha$, we have*

$$\mathrm{fat}_\alpha(\mathcal{F}) \leq \left( \frac{L_{\mathbf{hyp}} \cdot \mathrm{diam}(\mathcal{X})}{\alpha} \right)^{\mathrm{ddim}(\mathcal{X})}.$$

With the above upper bound, we provide the minimax expected regret for Lipschitz function classes explicitly in the next statement. Here, we consider the dimension $n$ and the diameter $\mathrm{diam}(\mathcal{X})$ as finite constants.

**Corollary C.4** (Upper bound, Lipschitz function). *Let $\mathcal{X} \subset \mathbb{R}^n$ be a metric space of finite diameter $\mathrm{diam}(\mathcal{X})$. For any function class $\mathcal{F} \subset [0,1]^{\mathcal{X}}$ of $L_{\mathbf{hyp}}$-Lipschitz function and any $L_{\mathbf{los}}$-Lipschitz and convex loss function $\ell$, the minimax expected regret of the transductive online regression satisfies*

$$R^{\mathbf{tr}}(T, \mathcal{F}) = \begin{cases} \tilde{\mathcal{O}}\left( L_{\mathbf{los}} \sqrt{L_{\mathbf{hyp}}} \cdot \sqrt{T} \right), & n = 1 \\ \tilde{\mathcal{O}}\left( L_{\mathbf{los}} L_{\mathbf{hyp}} \cdot \sqrt{T} \right), & n = 2 \\ \tilde{\mathcal{O}}\left( L_{\mathbf{los}} L_{\mathbf{hyp}} \cdot T^{\frac{n}{n+1}} \right), & n \geq 3 \end{cases} \cdot$$

*That is, the class of Lipschitz functions is transudctive online learnable.*

*Proof.* By Theorem 3.1, the minimax expected regret is upper bounded by

$$2 L_{\mathbf{los}} T \cdot \inf_{\alpha \geq 0} \left( 4\alpha + \frac{12}{\sqrt{T}} \int_\alpha^1 \sqrt{\mathrm{fat}_{\beta/4}(\mathcal{F}) \cdot c \log^2 \frac{T}{\beta}} \mathrm{d}\beta \right).$$

Then, since $\mathcal{F}$ is a class of $L_{\mathbf{hyp}}$-Lipschitz function, by Lemma C.3 we have

$$\mathrm{fat}_\alpha(\mathcal{F}) \leq \left( \frac{L_{\mathbf{hyp}} \cdot \mathrm{diam}(\mathcal{X})}{\alpha} \right)^{\mathrm{ddim}(\mathcal{X})},$$

where $\mathrm{ddim}\,(\mathcal{X}) \leq n$ is the doubling dimension for the metric space $\mathcal{X} \subset \mathbb{R}^n$. Then, for $n = 1$, taking $\alpha = \frac{1}{\sqrt{T}}$, the minimax expected regret is upper bounded by

$$\mathcal{O}\left(L_{\mathbf{los}}\right) \cdot \left(\sqrt{T} + \sqrt{TL_{\mathbf{hyp}}} \cdot \int_{1/\sqrt{T}}^1 \frac{1}{\beta^{1/2}} \cdot \sqrt{c}\log\frac{T}{\beta}\mathrm{d}\beta\right) = \tilde{\mathcal{O}}\left(L_{\mathbf{los}}\sqrt{L_{\mathbf{hyp}}} \cdot \sqrt{T}\right).$$

Similarly, for $n = 2$, taking $\alpha = \frac{1}{\sqrt{T}}$, the minimax expected regret is upper bounded by

$$\mathcal{O}\left(L_{\mathbf{los}}\right) \cdot \left(\sqrt{T} + L_{\mathbf{hyp}}\sqrt{T} \cdot \int_{1/\sqrt{T}}^1 \frac{1}{\beta} \cdot \sqrt{c}\log\frac{T}{\beta}\mathrm{d}\beta\right) = \tilde{\mathcal{O}}\left(L_{\mathbf{los}}L_{\mathbf{hyp}} \cdot \sqrt{T}\right).$$

Last, for $n > 2$, taking $\alpha = \frac{L_{\mathbf{hyp}}}{T^{1/n}}$, the minimax expected regret is upper bounded by

$$\mathcal{O}\left(L_{\mathbf{los}}\right) \cdot \left(\alpha T + \sqrt{T}L_{\mathbf{hyp}}^{\frac{n}{2}} \cdot \int_\alpha^1 \frac{1}{\beta^{\frac{n}{2}}} \cdot \sqrt{c}\log\frac{T}{\beta}\mathrm{d}\beta\right)$$
$$= \mathcal{O}\left(L_{\mathbf{los}}\right) \cdot \mathrm{polylog}(T) \cdot \left(\alpha T + \sqrt{T}L_{\mathbf{hyp}}^{\frac{n}{2}} \cdot \alpha^{1-\frac{n}{2}}\right) = \tilde{\mathcal{O}}\left(L_{\mathbf{los}}L_{\mathbf{hyp}} \cdot T^{\frac{n}{n+1}}\right).$$

Thus, we show the desired result. $\qquad\square$

The above result does not essentially give us a better rate for transductive online learning, since the sequential fat-shattering dimensions of Lipschitz classes are roughly equivalent to their fat-shattering dimensions. Next, we present the results for $k$-fold aggregations, whose sequential fat-shattering dimensions are infinite, yielding better rates for transductive online learning.

**$k$-fold aggregations.** We study the function class induced by $k$-fold aggregation, which is a mapping $G : \mathbb{R}^k \to [0,1]$. Given $k$ function classes $\mathcal{F}_1, \ldots, \mathcal{F}_k$ in $\mathbb{R}$, the function class defined by $G$ is

$$G(\mathcal{F}_1, \ldots, \mathcal{F}_k) := \{x \to G(F_1(x), \ldots, F_k(x)) : F_\kappa \in \mathcal{F}_\kappa, \forall \kappa \in [k]\}.$$

Let $\mathbf{e}$ be the all-one vector. The mapping $G : \mathbb{R}^k \to [0,1]$ commutes with shifts if

$$G(v) - r = G(v - r \cdot \mathbf{e}), \quad \forall\, v \in \mathbb{R}^k, r \in \mathbb{R}.$$

The above property is possessed by many natural aggregation mappings, including the maximum, minimum, median, and mean. The next statement provides an upper bound on the fat-shattering dimension of $k$-fold aggregations on general function classes.

**Theorem C.5** (See Theorem 1 in (Attias & Kontorovich, 2024)). *Given function classes $\mathcal{F}_1, \ldots, \mathcal{F}_k$, and an aggregation mapping $G$ that commutes with shifts, we have*

$$\mathrm{fat}_\alpha(G(\mathcal{F}_1, \ldots, \mathcal{F}_k)) \leq cd_\alpha \log^2 d_\alpha,$$

*where $d_\alpha = \sum_{\kappa \in [k]} \mathrm{fat}_\alpha(\mathcal{F}_\kappa)$ and $c$ is some universal constant.*

Now, we compute the minimax expected regret for $k$-fold aggregations in Lipschitz function classes. Here, we fix the range of function classes to $[0,1]$ for simplicity of calculation. Indeed, the results hold for all Lipschitz functions with bounded ranges by linear transformation.

**Corollary C.6** (Upper bound, $k$-fold aggregations). *Let $\mathcal{X} \subset \mathbb{R}^n$ be a metric space of finite diameter $\mathrm{diam}\,(\mathcal{X})$. For any bounded $L_{\mathbf{hyp}}$-Lipschitz function classes $\mathcal{F}_1, \ldots, \mathcal{F}_k \subset [0,1]^{\mathcal{X}}$, aggregation mapping $G$ that commutes with shifts, and any $L_{\mathbf{los}}$-Lipschitz and convex loss function $\ell$, the minimax expected regret of the transductive online regression for $G(\mathcal{F}_1, \ldots, \mathcal{F}_k)$ satisfies*

$$R^{\mathbf{tr}}(T, G(\mathcal{F}_1, \ldots, \mathcal{F}_k)) = \begin{cases} \tilde{\mathcal{O}}\left(\sqrt{k}L_{\mathbf{los}}\sqrt{L_{\mathbf{hyp}}} \cdot \sqrt{T}\right), & n = 1 \\ \tilde{\mathcal{O}}\left(\sqrt{k}L_{\mathbf{los}}L_{\mathbf{hyp}} \cdot \sqrt{T}\right), & n = 2 \\ \tilde{\mathcal{O}}\left(\sqrt{k}L_{\mathbf{los}}L_{\mathbf{hyp}} \cdot T^{\frac{n}{n+1}}\right), & n \geq 3 \end{cases}.$$

*That is, the class of $k$-fold aggregations on Lipschitz functions is transudctive online learnable.*

*Proof.* By Lemma C.3, we have for all $\kappa \in [k]$,

$$\mathrm{fat}_\alpha(\mathcal{F}_\kappa) \leq \left( \frac{L_{\mathbf{hyp}} \cdot \mathrm{diam}\,(\mathcal{X})}{\alpha} \right)^{\mathrm{ddim}(\mathcal{X})}.$$

In addition, by Theorem C.5 we have

$$\mathrm{fat}_\alpha(G(\mathcal{F}_1, \ldots, \mathcal{F}_k)) \leq \tilde{\mathcal{O}}\left( k \cdot \left( \frac{L_{\mathbf{hyp}}}{\alpha} \right)^{\mathrm{ddim}(\mathcal{X})} \right).$$

Therefore, the upper bounds can be computed using the same analysis as in Corollary C.4. □

**Functions with bounded variation.** We consider the class of functions with bounded variation. Specifically, let $\mathcal{F}^*$ be the set of all functions $f : [0,1] \to [0,1]$ with total variation of at most $V$. Here, for a $f \in \mathcal{F}^*$, we define its total variation as

$$\mathrm{TV}(f) = \sup_{p \in P} \sum_{i=0}^{n_{p-1}} |f(x_{i+1}) - f(x_i)|,$$

where the supremum is taken over the set $P = \{(x_0, \ldots, x_{n_p}), 0 \leq x_1 \leq \cdots \leq x_{n_p} \leq 1\}$ of all partitions of $[0,1]$. For a given parameter $\alpha > 0$, we define the metric covering number $\mathcal{N}(\mathcal{F}^*, \alpha, \mu)$ as the smallest number of sets of radius $\alpha$ under metric $\mu$ whose union contains $\mathcal{F}^*$. We remark that this definition is different from our previous definition of a $\ell_p$-covering number on a sequence of example $x$. We introduce it to bound the fat-shattering dimension of the class of bounded variation. Now, we investigate the covering number under $L_1(\mathrm{d}\mathcal{P})$ metrics, where $\mathcal{P}$ is a probability distribution on $[0,1]$. The following statement provides an upper bound.

**Theorem C.7** (See Theorem 1 in (Bartlett et al., 2006)). *Let $\mathcal{F}^*$ be the set of all functions $f : [0,1] \to [0,1]$ with total variation of at most $V$, we have*

$$\sup_{\mathcal{P}} \log_2 \mathcal{N}(\mathcal{F}^*, \alpha, L_1(\mathrm{d}\mathcal{P})) = \frac{12V}{\alpha}.$$

The next statement upper bounds the fat-shattering dimension of $\mathcal{F}^*$ by the covering number.

**Theorem C.8** (See Theorem 2 in (Bartlett et al., 2006)). *Let $\mathcal{F}^*$ be the set of all functions $f : [0,1] \to [0,1]$ with total variation of at most $V$, we have*

$$\mathrm{fat}_{4\alpha}(\mathcal{F}^*) \leq 32 \cdot \sup_{\mathcal{P}} \log_2 \mathcal{N}(\mathcal{F}^*, \alpha, L_1(\mathrm{d}\mathcal{P})).$$

Note that the fat-shattering dimension of class of functions with bounded variation satisfies $\mathrm{fat}_\alpha(\mathcal{F}^*) = \mathcal{O}\left( \frac{\mathrm{TV}(f)}{\alpha} \right)$, leading to the minimax expected regret in the following theorem.

**Corollary C.9** (Upper bound, bounded variation). *Let $\mathcal{F}^*$ be the set of all functions $f : [0,1] \to [0,1]$ with total variation of at most $V$. Let $\ell$ be a $L_{\mathbf{los}}$-Lipschitz loss function. The minimax expected regret of the transductive online regression for $\mathcal{F}^*$ under loss $\ell$ satisfies $R^{\mathbf{tr}}(T, \mathcal{F}^*) = \tilde{\mathcal{O}}\left( L_{\mathbf{los}} \cdot \sqrt{VT} \right)$. That is, the class of functions with bounded variation on $[0,1]$ is transudctive online learnable.*

*Proof.* Combining Theorem C.7 and Theorem C.8, we have that the fat-shattering dimension of $\mathcal{F}^*$ satisfies

$$\mathrm{fat}_\alpha(\mathcal{F}^*) = \mathcal{O}\left( \frac{V}{\alpha} \right).$$

By Theorem 3.1, the minimax expected regret is upper bounded by

$$2L_{\mathbf{los}}T \cdot \inf_{\alpha \geq 0} \left( 4\alpha + \frac{12}{\sqrt{T}} \int_\alpha^1 \sqrt{\mathrm{fat}_{\beta/4}(\mathcal{F}) \cdot c \log^2 \frac{T}{\beta}} \mathrm{d}\beta \right).$$

Then, taking $\alpha = \frac{1}{\sqrt{T}}$, the minimax expected regret is upper bounded by

$$\mathcal{O}\left( L_{\mathbf{los}} \right) \cdot \left( \sqrt{T} + \sqrt{VT} \cdot \int_{1/\sqrt{T}}^1 \frac{1}{\beta^{1/2}} \cdot \sqrt{c} \log \frac{T}{\beta} \mathrm{d}\beta \right) = \tilde{\mathcal{O}}\left( L_{\mathbf{los}} \cdot \sqrt{VT} \right).$$

□

# D   MINIMAX REGRET FOR ONLINE REGRESSION WITH PREDICTIONS

In this section, we present the algorithms and missing proofs in Section 4.

## D.1   ONLINE LEARNER UNDER ZERO-ONE METRIC

In this section, we quantify the performance of a Predictor $\mathcal{P}$ as the expected number of mistakes that $\mathcal{P}$ makes, which is

$$M_{\mathcal{P}}(x_{1:T}) := \mathbb{E}\left[\sum_{t=2}^{T} \mathbf{1}_{\mathcal{P}(x_{1:t-1})_t \neq x_t}\right],$$

where we use $\mathcal{P}(x_{1:t-1})_{1:T}$ to denote its predictions $\widehat{x_{1:T}}$ given the previous examples $x_{1:t-1}$, and the expectation is taken only over the randomness of $\mathcal{P}$. We make assumptions about the consistency and the laziness of the Predictor $\mathcal{P}$ (c.f, Section 2.2 in (Raman & Tewari, 2024)), which are defined below.

**Definition D.1** (Consistency)**.** *For every sequence $x_{1:T} \in \mathcal{X}^T$ and for each time $t \in [T]$, $\mathcal{P}$ is consistent if its prediction $\widehat{x_{1:T}}^t$ satisfies $\mathcal{P}(x_{1:t})_{1:t} = x_{1:t}$.*

The assumption about consistency is natural, since we can hard code the prediction of $x_{1:t}$ to be the input. Next, we introduce the definition of laziness.

**Definition D.2** (Laziness)**.** *$\mathcal{P}$ is consistent if its prediction satisfies the following property. For every sequence $x_{1:T} \in \mathcal{X}^T$ and for each time $t \in [T]$, if $\mathcal{P}(x_{1:t-1})_t = x_t$, then $\mathcal{P}(x_{1:t}) = \mathcal{P}(x_{1:t-1})$. That is, $\mathcal{P}$ does not change its prediction if it is correct.*

The assumption about laziness is also mild, since non-lazy online Predictors can be converted into lazy ones (Littlestone, 1989). Next, we introduce our online learner given $\mathcal{P}$, whose intuition follows from (Raman & Tewari, 2024). Suppose that $\mathcal{P}$ makes mistakes at times $t_1, \dots t_c \in [T]$, due to the assumption of laziness and consistency, the predictions of $\mathcal{P}$ between $t_j$ and $t_{j+1}$ are correct and unchanged for all $j \in [c]$. Thus, whenever we detect a mistake, we notify $\mathcal{P}$ and retrieve its new sequence of predictions. Next, we initialize a new transductive online learner $\mathcal{B}$ with predicted future inputs given by $\mathcal{P}$ and report its predicted label $\widehat{y}_t$ until the next time $\mathcal{P}$ makes a mistake. This gives an error rate of roughly $M_{\mathcal{P}}(x_{1:T}) \cdot R_{\mathcal{B}}^{\mathbf{tr}}(T, \mathcal{F})$. Our algorithm is presented in Algorithm 2.

---

**Algorithm 2** Online Learner with Prediction

1: **Input:** Function class $\mathcal{F}$, transductive online learner $\mathcal{B}$, Predictor $\mathcal{P}$, time interval $[T]$, sequence of examples and labels $(x, y)_{1:T}$ revealed by the adversary sequentially
2: **Output:** to $y_{1:T}$
3: $i \leftarrow 0$
4: **for** $t \in [T]$ **do**
5:     $\mathcal{P}$ makes prediction $\mathcal{P}(x_{1:t})$ such that $\mathcal{P}(x_{1:t})_{1:t} = x_{1:t}$
6:     **if** $t = 1$ or $\mathcal{P}(x_{1:t})_{t+1} \neq x_{t+1}$ (i.e. $\mathcal{P}$ makes a mistake) **then**
7:         $i \leftarrow i + 1$
8:         Run a new transductive online learner $\mathcal{B}^i$ initialized with the sequence $\mathcal{P}(x_{1:t+1})_{t+1:T}$
9:     **end if**
10:    **Return:** Prediction $\widehat{y}_t$ by the current transductive online learner
11:    Reveal the actual label $y_t$ and input into the current transductive online learner
12: **end for**

---

The next statement upper bounds the expected error of Algorithm 2.

**Lemma D.3** (Analogous to Lemma 20 in (Raman & Tewari, 2024))**.** *Given a Predictor $\mathcal{P}$ and an transductive online learner $\mathcal{B}$, for any function class $\mathcal{F} \subset \mathcal{Y}^{\mathcal{X}}$, loss function $\ell$, and data stream $(x_1, y_1), \dots, (x_T, y_T)$, the minimax expected regret of Algorithm 2 is bounded by $(M_{\mathcal{P}}(x_{1:T}) + 1)R_{\mathcal{B}}^{\mathbf{tr}}(T, \mathcal{F})$.*

*Proof.* The proof is similar to (Raman & Tewari, 2024), we keep it here for completeness. Let $\mathcal{A}$ be the learner in Algorithm 2. Let $c$ be the random variable that denotes the total number of mistakes

made by $\mathcal{P}$, and let $t_1, \ldots, t_c$ be the random time points at which these errors occur. Without loss of generality, we assume $c > 0$, since otherwise, due to the consistency and laziness of $\mathcal{P}$ (see Definition D.1 and Definition D.2), $\mathcal{P}(x_{1:1}) = x_{1:T}$ for every $t \in [T]$. Thus, we only run one transductive online learner $\mathcal{B}^1$, and so the regret is at most $R_{\mathcal{B}}^{\mathbf{tr}}(T, \mathcal{F})$.

Now, we partition the sequence of time points into disjoint intervals $(t_0, \ldots, t_1 - 1), (t_1, \ldots, t_2 - 1), \ldots, (t_c, \ldots, t_{c+1} - 1)$, where $t_0 := 1$ and $t_{c+1} - 1 := T$. Fix an arbitrary $i \in [c]$. Due to our algorithm construction, for each $j \in \{t_i, \ldots, t_{i+1} - 1\}$, we have $\mathcal{P}(x_{1:j})_{1:t_{i+1}-1} = x_{1:t_{i+1}-1}$. Thus, the transductive online learner $\mathcal{B}_i$ is applied in the example stream

$$x_{t_i}, \ldots, x_{t_{i+1}-1}, \mathcal{P}(x_{1:t_i})_{t_{i+1}}, \ldots, \mathcal{P}(x_{1:t_i})_{t_T}.$$

Let $h^i \in \operatorname{argmin}_{f \in \mathcal{F}} \sum_{t=t_i}^{t_{i+1}-1} \ell(f(x_t), y_t)$ be an optimal function for duration $(t_i, \ldots, t_{i+1} - 1)$. Let $y_t^i = y_t$ for all $t_i \le t \le t_{i+1} - 1$ and $y_t^i = h^i(\mathcal{P}(x_{1:t_i})_t)$ for all $t \ge t_{i+1}$. Then, we observe that

$$\inf_{f \in \mathcal{F}} \sum_{t_i}^{T} \ell(f(\mathcal{P}(x_{1:t_i})_t), y_t^i) = \sum_{t=t_i}^{t_{i+1}-1} \ell(h^i(x_t), y_t) = \inf_{f \in \mathcal{F}} \sum_{t=t_i}^{t_{i+1}-1} \ell(f(x_t), y_t).$$

Next, we consider the hypothetical labeled stream

$$S = (x_{t_i}, y_{t_i}^i), \ldots, (x_{t_{i+1}-1}, y_{t_{i+1}-1}^i), (\mathcal{P}(x_{1:t_i})_{t_{i+1}}, y_{t_{i+1}}^i) \ldots, (\mathcal{P}(x_{1:t_i})_{t_T}, y_T^i)$$

Then, from the definition of the minimax expected regret $R_{\mathcal{B}}^{\mathbf{tr}}(T, \mathcal{F})$, the expected loss $\mathcal{B}^i$ has in the stream $S$ is at most

$$R_{\mathcal{B}}^{\mathbf{tr}}(T - t_i + 1, \mathcal{F}) + \inf_{f \in \mathcal{F}} \sum_{t_i}^{T} \ell(f(\mathcal{P}(x_{1:t_i})_t), y_t^i) = R_{\mathcal{B}}^{\mathbf{tr}}(T - t_i + 1, \mathcal{F}) + \inf_{f \in \mathcal{F}} \sum_{t=t_i}^{t_{i+1}-1} \ell(f(x_t), y_t).$$

Thus, $\mathcal{A}$ has loss at most $R_{\mathcal{B}}^{\mathbf{tr}}(T, \mathcal{F}) + \inf_{f \in \mathcal{F}} \sum_{t=t_i}^{t_{i+1}-1} \ell(f(x_t), y_t)$ during $(t_i, t_{i+1} - 1)$ in expectation. Then, we have

$$\mathbb{E}\left[\sum_{t=1}^{T} \ell(\mathcal{A}_t, h^*(x_t))\right] = \sum_{i=0}^{c} \left(\mathbb{E}\left[\sum_{t=t_i}^{t_{i+1}-1} \ell(\mathcal{A}_t, h^*(x_t))\right]\right)$$

$$\le \sum_{i=0}^{c} \left(R_{\mathcal{B}}^{\mathbf{tr}}(T, \mathcal{F}) + \inf_{f \in \mathcal{F}} \sum_{t=t_i}^{t_{i+1}-1} \ell(f(x_t), y_t)\right)$$

$$\le (c+1) R_{\mathcal{B}}^{\mathbf{tr}}(T, \mathcal{F}) + \inf_{f \in \mathcal{F}} \sum_{t=1}^{T} \ell(f(x_t), y_t),$$

where the expectation is only on the randomness of each $\mathcal{B}^i$. Last, since $\mathbb{E}[c] = M_{\mathcal{P}}(x_{1:T})$, taking an outer expectation of the randomness of $\mathcal{P}$, we show that the minimax expected regret of $\mathcal{A}$ is at most $(M_{\mathcal{P}}(x_{1:T}) + 1) R_{\mathcal{B}}^{\mathbf{tr}}(T, \mathcal{F})$. □

A drawback of the above error bound is that, when $M_{\mathcal{P}}(x_{1:T})$ is large (e.g., $\Omega(\sqrt{T})$), the upper bound is suboptimal. To overcome this, we partition the time duration $[T]$ to $c$ equi-distant intervals and run a fresh copy of Algorithm 2 for each interval. Then we run MWA using experts with all $c \in [T-1]$ as inputs. We show that the minimax expected regret for each expert with input $c$ is roughly $(M_{\mathcal{P}}(x_{1:T}) + c) \cdot \bar{R}_{\mathcal{B}}^{\mathbf{tr}}\left(\frac{T}{c}, \mathcal{F}\right)$, thus MWA gives our desired error bound that has $M_{\mathcal{P}}(x_{1:T})$ as an interpolation factor. We present the algorithm for each expert in Algorithm 3 and MWA in Algorithm 4. Before introducing the error bound, we state a lemma that upper bounds a positive sublinear function with countable domain by a concave sublinear function. We use this result to upper bound the transductive online regret $R_{\mathcal{B}}^{\mathbf{tr}}(T, \mathcal{F})$ by a concave sublinear function $\bar{R}_{\mathcal{B}}^{\mathbf{tr}}(T, \mathcal{F})$.

**Lemma D.4** (see Lemma 5.17 in (Ceccherini-Silberstein et al., 2017)). *Let $g : \mathbb{Z}_+ \to \mathbb{R}_+$ be a positive sublinear function. Then $g$ is bounded from above by a concave sublinear function $\bar{g} : \mathbb{R}_+ \to \mathbb{R}_+$.*

Next, we compute the minimax expected regret of Algorithm 4.

---

**Algorithm 3** Expert($c$)

---

1: **Input:** Learner $\mathcal{A}$ in Algorithm 2, number of pieces $c$, function class $\mathcal{F}$, time interval $[T]$, sequence of examples and labels $(x, y)_{1:T}$ revealed by the adversary sequentially
2: **Output:** Predictions to $y_{1:T}$
3: Let $\tilde{t}_j = j \left\lceil \frac{T}{c+1} \right\rceil$ for each $j \in [c], \tilde{t}_0 = 0$, and $\tilde{t}_{c+1} = T$
4: Obtain independent learner $\mathcal{A}_j$ from Algorithm 2 for each $j \in [c]$
5: $j \leftarrow 0$
6: **for** $t \in [T]$ **do**
7:     **if** $t = \tilde{t}_j + 1$ **then**
8:         $j \leftarrow j + 1$
9:         Run a new instance $\mathcal{A}_j$ initialized with time duration $[\tilde{t}_j + 1, \tilde{t}_{j+1}]$ {The Predictor $\mathcal{P}$ in $\mathcal{A}_j$ predicts the restricted sequence $x_{\tilde{t}_j+1:\tilde{t}_{j+1}}$}
10:     **end if**
11:     **Return:** Prediction $\widehat{y}_t$ by $\mathcal{A}_j$
12:     Reveal the actual label $y_t$ from the adversary and input into $\mathcal{A}_j$
13: **end for**

---

**Algorithm 4** Online Learner with Prediction

---

1: **input:** Function class $\mathcal{F}$, time interval $[T]$, sequence of examples and labels $(x, y)_{1:T}$ revealed by the adversary sequentially
2: **Output:** Predictions to $y_{1:T}$
3: For each $c \in [T-1]$, let Expert($c$) denote an instance of Algorithm 3 with input $c$
4: Obtain the prediction from MWA (see Theorem C.1) using $\{\text{Expert}(c)\}_{c \in [T-1]}$ over $(x, y)_{1:T}$

---

**Lemma D.5** (Analogous to bound(ii) in Theorem 16 in (Raman & Tewari, 2024)). *Given a Predictor $\mathcal{P}$ and an transductive online learner $\mathcal{B}$, for any function class $\mathcal{F} \subset \mathcal{Y}^{\mathcal{X}}$, Lipschitz and convex loss function $\ell$, and data stream $(x_1, y_1), \ldots, (x_T, y_T)$, the minimax expected regret of Algorithm 4 is bounded by*

$$2(M_{\mathcal{P}}(x_{1:T}) + 1)\bar{R}^{\mathbf{tr}}_{\mathcal{B}} \left( \frac{T}{M_{\mathcal{P}}(x_{1:T}) + 1} + 1, \mathcal{F} \right) + \sqrt{T \log T}.$$

*Proof.* We note that it suffices to show that the minimax expected regret of the expert $c$ is at most $(M_{\mathcal{P}}(x_{1:T}) + c + 1)\bar{R}^{\mathbf{tr}}_{\mathcal{B}} \left( \frac{T}{c+1} + 1, \mathcal{F} \right)$ for every $c \in [T-1]$, then by the guarantee of MWA (see Theorem C.1), we have our desired upper bound taking $c = \lceil M_{\mathcal{P}}(x_{1:T}) \rceil$.

Now, we fix a $c \in [T-1]$. Let $\tilde{t}_j = j \left\lceil \frac{T}{c+1} \right\rceil$ for each $j \in [c], \tilde{t}_0 = 0$, and $\tilde{t}_{c+1} = T$. Let $\mathcal{E}$ denote the expert with input $c$ in Algorithm 3, then we have

$$\mathbb{E}\left[ \sum_{i=1}^{T} \ell(\mathcal{E}(x_t), y_t) \right] = \mathbb{E}\left[ \sum_{j=0}^{c} \sum_{t=\tilde{t}_j+1}^{\tilde{t}_{j+1}} \ell(\mathcal{A}^j(x_t), y_t) \right],$$

where $\mathcal{A}^j$ is the learner with time duration $[\tilde{t}_j + 1, \tilde{t}_{j+1}]$. Let $m_i$ be the number of mistakes that Predictor $\mathcal{P}$ makes in $\mathcal{A}^j$. Then, by the bound in Lemma D.3, we have

$$\mathbb{E}\left[ \sum_{i=1}^{T} \ell(\mathcal{E}(x_t), y_t) \right] \leq \mathbb{E}\left[ \sum_{j=1}^{c} (m_j + 1)\bar{R}^{\mathbf{tr}}_{\mathcal{B}}(\tilde{t}_{j+1} - \tilde{t}_j, \mathcal{F}) + \inf_{f \in \mathcal{F}} \sum_{t=\tilde{t}_j+1}^{\tilde{t}_{j+1}} \ell(f(x_t), y_t) \right]$$

$$\leq \mathbb{E}\left[ \sum_{j=1}^{c} (m_j + 1)\bar{R}^{\mathbf{tr}}_{\mathcal{B}}(\tilde{t}_{j+1} - \tilde{t}_j, \mathcal{F}) \right] + \inf_{f \in \mathcal{F}} \sum_{t=1}^{T} \ell(f(x_t), y_t).$$

Then, it suffices to bound the first term $\mathbb{E}\left[ \sum_{j=1}^{c} (m_j + 1)\bar{R}^{\mathbf{tr}}_{\mathcal{B}}(\tilde{t}_{j+1} - \tilde{t}_j, \mathcal{F}) \right]$ by $(M_{\mathcal{P}}(x_{1:T}) + c + 1)\bar{R}^{\mathbf{tr}}_{\mathcal{B}}(\frac{T}{c+1} + 1, \mathcal{F})$. Note that $\bar{R}^{\mathbf{tr}}_{\mathcal{B}}(T, \mathcal{F})$ is a concave function in $T$ by our construction, then by

Jensen's inequality, we have

$$\mathbb{E}\left[\sum_{j=1}^{c}(m_j+1)\bar{R}_{\mathcal{B}}^{\mathbf{tr}}(\tilde{t}_{j+1}-\tilde{t}_j,\mathcal{F})\right] \leq \mathbb{E}\left[\left(\sum_{j=1}^{c}(m_j+1)\right)\cdot\bar{R}_{\mathcal{B}}^{\mathbf{tr}}\left(\frac{\sum_{j=1}^{c}(m_j+1)(\tilde{t}_{j+1}-\tilde{t}_j)}{\sum_{j=1}^{c}(m_j+1)},\mathcal{F}\right)\right].$$

Let $M = \sum_{j=1}^{c}m_j$ such that $\mathbb{E}[M] = M_{\mathcal{P}}(x_{1:T})$, then we have

$$\frac{\sum_{j=1}^{c}(m_j+1)(\tilde{t}_{j+1}-\tilde{t}_j)}{\sum_{j=1}^{c}(m_j+1)} = \frac{T+\sum_{j=1}^{c}(m_j)(\tilde{t}_{j+1}-\tilde{t}_j)}{M+c+1} = \frac{T+\sum_{j=1}^{c}m_j\cdot\left\lceil\frac{T}{c+1}\right\rceil}{M+c+1},$$

where the last step follows from our definition of $\tilde{t}_j = \left\lceil\frac{T}{c+1}\right\rceil$. Then, we have

$$\frac{\sum_{j=1}^{c}(m_j+1)(\tilde{t}_{j+1}-\tilde{t}_j)}{\sum_{j=1}^{c}(m_j+1)} = \frac{T+M\cdot\left\lceil\frac{T}{c+1}\right\rceil}{M+c+1} \leq \frac{T}{c+1}+1.$$

Therefore, we have

$$\mathbb{E}\left[\sum_{j=1}^{c}(m_j+1)\bar{R}_{\mathcal{B}}^{\mathbf{tr}}(\tilde{t}_{j+1}-\tilde{t}_j,\mathcal{F})\right] \leq \mathbb{E}\left[(M+c+1)\cdot\bar{R}_{\mathcal{B}}^{\mathbf{tr}}\left(\frac{T}{c+1}+1,\mathcal{F}\right)\right]$$

$$= (M_{\mathcal{P}}(x_{1:T})+c+1)\cdot\bar{R}_{\mathcal{B}}^{\mathbf{tr}}\left(\frac{T}{c+1}+1,\mathcal{F}\right).$$

This proves our desired bound. □

## D.2 ONLINE LEARNER UNDER $\varepsilon$-BALL METRIC

In this section, we quantify the performance of a Predictor $\mathcal{P}$ as the expected number of times that its prediction $\mathcal{P}$ is outside the $\varepsilon$-ball of the real input $x_t$. Consider a metric space $(\mathcal{X}, \mathrm{d})$ of examples, the $\varepsilon$-ball of a $x \in \mathcal{X}$ is $B(x) := \{x' \in \mathcal{X}, \mathrm{d}(x', x) < \varepsilon\}$. Then, our $\varepsilon$-ball metric for the Predictor is defined as

$$M_{\mathcal{P}}(\varepsilon, x_{1:T}) := \mathbb{E}\left[\sum_{t=2}^{T}\mathbf{1}_{\mathrm{d}(\mathcal{P}(x_{1:t-1})_t, x_t)\geq\varepsilon}\right],$$

where the expectation is taken only over the randomness of $\mathcal{P}$. We extend the notion of laziness from the previous sections in the sense of $\varepsilon$-ball.

**Definition D.6** (Laziness). *$\mathcal{P}$ is consistent if its prediction satisfies the following property. For every sequence $x_{1:T} \in \mathcal{X}^T$ and for each time $t \in [T]$, if $\mathrm{d}(\mathcal{P}(x_{1:t-1})_t, x_t) \leq \varepsilon$, then $\mathcal{P}(x_{1:t}) = \mathcal{P}(x_{1:t-1})$. That is, $\mathcal{P}$ does not change its prediction if it is inside the $\varepsilon$-ball.*

We extend Algorithm 2 to the new notion of predictability. Suppose that the prediction is outside the $\varepsilon$-ball at times $t_1, \ldots t_c \in [T]$, then we run a separate transductive online learner $\mathcal{B}$ for each duration $t_j, \ldots, t_{j+1}$ for $j \in [c]$, i.e., we start a new instance whenever the prediction is outside the $\varepsilon$-ball. Since the prediction is always inside the $\varepsilon$-ball between $t_j$ and $t_{j+1}$, then if the function class is $L$-Lipschitz, our error bound has an additional $\varepsilon LT$ factor. We present this algorithm in Algorithm 5.

We upper bound the minimax expected regret of Algorithm 5 in the next lemma.

**Lemma D.7.** *Given a Predictor $\mathcal{P}$ and an transductive online learner $\mathcal{B}$, for any function class $\mathcal{F} \subset \mathcal{Y}^{\mathcal{X}}$ of $L_{\mathbf{hyp}}$-Lipschitz function, $L_{\mathbf{los}}$-Lipschitz loss function $\ell$, and data stream $(x_1, y_1), \ldots, (x_T, y_T)$, the minimax expected regret of Algorithm 5 is bounded by $(M_{\mathcal{P}}(\varepsilon, x_{1:T}) + 1)R_{\mathcal{B}}^{\mathbf{tr}}(T, \mathcal{F}) + \varepsilon L_{\mathbf{los}}L_{\mathbf{hyp}} \cdot T$.*

*Proof.* This proof is extended from Lemma D.3. Let $\mathcal{A}$ be the learner in Algorithm 5. Let $c$ be the random variable denoting the total number of times that the prediction is outside the $\varepsilon$-ball, and let $t_1, \ldots, t_c$ be the random time points at which these errors occur.

---

**Algorithm 5** Online Learner with Prediction

---

1: **Input:** function class $\mathcal{F}$, transductive online learner $\mathcal{B}$, Predictor $\mathcal{P}$, time interval $[T]$, sequence of examples and labels $(x, y)_{1:T}$ revealed by the adversary sequentially
2: **Output:** Predictions to $y_{1:T}$
3: $i \leftarrow 0$
4: **for** $t \in [T]$ **do**
5:    $\mathcal{P}$ makes prediction $\mathcal{P}(x_{1:t})$ such that $\mathrm{d}(\mathcal{P}(x_{1:t})_l, x_l) < \varepsilon$ for each $l \in [t]$
6:    **if** $t = 1$ or $\mathrm{d}(\mathcal{P}(x_{1:t})_{t+1}, x_{t+1}) \geq \varepsilon$ (i.e. the prediction is outside the $\varepsilon$-ball) **then**
7:       $i \leftarrow i + 1$
8:       Run a new transductive online learner $\mathcal{B}^i$ initialized with the sequence $\mathcal{P}(x_{1:t+1})_{t+1:T}$
9:    **end if**
10:   **Return:** Prediction $\widehat{y}_t$ by the current transductive online learner
11:   Reveal the actual label $y_t$ and input into the current transductive online learner
12: **end for**

---

First, we consider the case that $c = 0$, then due to the laziness of $\mathcal{P}$ (see Definition D.6), $|\mathcal{P}(x_{1:1})_t - x_t| \leq \varepsilon$ for every $t \in [T]$. Thus, we only run one transductive online learner $\mathcal{B}^1$, and so we have

$$\mathbb{E}\left[\sum_{t=1}^{T} \ell(\mathcal{A}_t, y_t)\right] = \mathbb{E}\left[\sum_{t=1}^{T} \ell(\mathcal{B}^1(\mathcal{P}(x_{1:1})_t), y_t)\right] \leq \inf_{f \in \mathcal{F}} \left(\sum_{t=1}^{T} \ell(f(\mathcal{P}(x_{1:1})_t), y_t)\right) + R_{\mathcal{B}}^{\mathbf{tr}}(T, \mathcal{F}).$$

Since we assume that $\mathcal{F}$ is a class of $L_{\mathbf{hyp}}$-Lipschitz function, we have for each $f \in \mathcal{F}$ and $t \in [T]$, $|f(\mathcal{P}(x_{1:1})_t) - f(x_t)| \leq \varepsilon L_{\mathbf{hyp}}$. Additionally, since we also assume that the loss function is $L_{\mathbf{los}}$-Lipschitz, we have for each $f \in \mathcal{F}$ and $t \in [T]$, $|\ell(f(\mathcal{P}(x_{1:1})_t), y_t) - \ell(f(x_t), y_t)| \leq \varepsilon L_{\mathbf{los}} L_{\mathbf{hyp}}$. Therefore, we have

$$\inf_{f \in \mathcal{F}} \left(\sum_{t=1}^{T} \ell(f(\mathcal{P}(x_{1:1})_t), y_t)\right) \leq \inf_{f \in \mathcal{F}} \left(\sum_{t=1}^{T} \ell(f(x_t), y_t)\right) + \varepsilon L_{\mathbf{los}} L_{\mathbf{hyp}} \cdot T.$$

Thus, the minimax expected regret of $\mathcal{A}$ is at most $R_{\mathcal{B}}^{\mathbf{tr}}(T, \mathcal{F}) + \varepsilon L_{\mathbf{los}} L_{\mathbf{hyp}} \cdot T$.

Next, we consider the case that $c > 0$. We partition the sequence of time points into disjoint intervals $(t_0, \dots, t_1 - 1), (t_1, \dots, t_2 - 1), \dots, (t_c, \dots, t_{c+1} - 1)$, where $t_0 := 1$ and $t_{c+1} - 1 := T$. Fix an arbitrary $i \in [c]$. By our algorithm construction, the transductive online learner $\mathcal{B}_i$ is applied in the example stream $\mathcal{P}(x_{1:t_i})_{t_i}, \dots, \mathcal{P}(x_{1:t_i})_{t_T}$. Let $h^i \in \arg\min_{f \in \mathcal{F}} \sum_{t=t_i}^{t_{i+1}-1} \ell(f(\mathcal{P}(x_{1:t_i})_t), y_t)$ be an optimal function for duration $(t_i, \dots, t_{i+1} - 1)$. Let $y_t^i = y_t$ for all $t_i \leq t \leq t_{i+1} - 1$ and $y_t^i = h^i(\mathcal{P}(x_{1:t_i})_t)$ for all $t \geq t_{i+1}$. Then we observe that

$$\inf_{f \in \mathcal{F}} \sum_{t_i}^{T} \ell(f(\mathcal{P}(x_{1:t_i})_t), y_t^i) = \sum_{t=t_i}^{t_{i+1}-1} \ell(h^i(\mathcal{P}(x_{1:t_i})_t), y_t) = \inf_{f \in \mathcal{F}} \sum_{t=t_i}^{t_{i+1}-1} \ell(f(\mathcal{P}(x_{1:t_i})_t), y_t).$$

Next, we consider the hypothetical labeled stream

$$\mathcal{S} = (\mathcal{P}(x_{1:t_i})_{t_i}, y_{t_{i+1}}^i) \dots, (\mathcal{P}(x_{1:t_i})_{t_T}, y_T^i)$$

Then, from the definition of the minimax expected regret $R_{\mathcal{B}}^{\mathbf{tr}}(T, \mathcal{F})$, the expected loss $\mathcal{B}^i$ has in the stream $S$ is at most

$$R_{\mathcal{B}}^{\mathbf{tr}}(T - t_i + 1, \mathcal{F}) + \inf_{f \in \mathcal{F}} \sum_{t_i}^{T} \ell(f(\mathcal{P}(x_{1:t_i})_t), y_t^i) = R_{\mathcal{B}}^{\mathbf{tr}}(T - t_i + 1, \mathcal{F}) + \inf_{f \in \mathcal{F}} \sum_{t=t_i}^{t_{i+1}-1} \ell(f(\mathcal{P}(x_{1:t_i})_t), y_t).$$

Now, since we use the same Predictor $\mathcal{P}$ during $(t_i, t_{i+1} - 1)$, which means that $\mathrm{d}(\mathcal{P}(x_{1:t_i})_t, x_t) \leq \varepsilon$ for every $t \in (t_i, t_{i+1} - 1)$. Then, by a similar Lipschitz argument, we have

$$\inf_{f \in \mathcal{F}} \left(\sum_{t=t_i}^{t_{i+1}-1} \ell(f(\mathcal{P}(x_{1:t_i})_t), y_t)\right) \leq \inf_{f \in \mathcal{F}} \left(\sum_{t=t_i}^{t_{i+1}-1} \ell(f(x_t), y_t)\right) + \varepsilon L_{\mathbf{los}} L_{\mathbf{hyp}} \cdot (t_{i+1} - t_i).$$

Therefore, $\mathcal{A}$ has loss at most $R_{\mathcal{B}}^{\mathbf{tr}}(T, \mathcal{F}) + \inf_{f \in \mathcal{F}} \left( \sum_{t=t_i}^{t_{i+1}-1} \ell(f(x_t), y_t) \right) + \varepsilon L_{\mathbf{los}} L_{\mathbf{hyp}} \cdot (t_{i+1} - t_i)$ during $(t_i, t_{i+1} - 1)$ in expectation. Then, we have

$$
\begin{aligned}
\mathbb{E}\left[ \sum_{t=1}^{T} \ell(\mathcal{A}_t, h^*(x_t)) \right] &= \sum_{i=0}^{c} \left( \mathbb{E}\left[ \sum_{t=t_i}^{t_{i+1}-1} \ell(\mathcal{A}_t, h^*(x_t)) \right] \right) \\
&\leq \sum_{i=0}^{c} \left( R_{\mathcal{B}}^{\mathbf{tr}}(T, \mathcal{F}) + \inf_{f \in \mathcal{F}} \left( \sum_{t=t_i}^{t_{i+1}-1} \ell(f(x_t), y_t) \right) + \varepsilon L_{\mathbf{los}} L_{\mathbf{hyp}} \cdot (t_{i+1} - t_i) \right) \\
&\leq (c+1) R_{\mathcal{B}}^{\mathbf{tr}}(T, \mathcal{F}) + \inf_{f \in \mathcal{F}} \left( \sum_{t=1}^{T} \ell(f(x_t), y_t) \right) + \varepsilon L_{\mathbf{los}} L_{\mathbf{hyp}} T,
\end{aligned}
$$

where the expectation is only on the randomness of each $\mathcal{B}^i$. Last, since $\mathbb{E}[c] = M_{\mathcal{P}}(\varepsilon, x_{1:T})$, taking an outer expectation of the randomness of $\mathcal{P}$, we show that the minimax expected regret of $\mathcal{A}$ is at most $(M_{\mathcal{P}}(\varepsilon, x_{1:T}) + 1) R_{\mathcal{B}}^{\mathbf{tr}}(T, \mathcal{F}) + \varepsilon L_{\mathbf{los}} L_{\mathbf{hyp}} T$. □

Next, we extend Algorithm 4 under the notion of $\varepsilon$-ball metric, where we construct each expert by the subroutine in Algorithm 5. The algorithm is presented in Algorithm 6.

---

**Algorithm 6** Online Learner with Prediction

1: **Input:** function class $\mathcal{F}$, time interval $[T]$, sequence of examples and labels $(x, y)_{1:T}$ revealed by the adversary sequentially
2: **Output:** Predictions to $y_{1:T}$
3: For each $c \in [T-1]$, let Expert($c$) denote an instance of Algorithm 3 using the online learner in Algorithm 5
4: Obtain the prediction from MWA (see Theorem C.1) using $\{\text{Expert}(c)\}_{c \in [T-1]}$ over $(x, y)_{1:T}$

---

The following statement bounds the minimax expected regret of Algorithm 6.

**Lemma D.8.** *Given a Predictor $\mathcal{P}$ and an transductive online learner $\mathcal{B}$, for any function class $\mathcal{F} \subset \mathcal{Y}^{\mathcal{X}}$ of $L_{\mathbf{hyp}}$-Lipschitz function, $L_{\mathbf{los}}$-Lipschitz loss function $\ell$, and data stream $(x_1, y_1), \ldots, (x_T, y_T)$, the minimax expected regret of Algorithm 6 is bounded by*

$$
2(M_{\mathcal{P}}(\varepsilon, x_{1:T}) + 1) \bar{R}_{\mathcal{B}}^{\mathbf{tr}} \left( \frac{T}{M_{\mathcal{P}}(\varepsilon, x_{1:T}) + 1} + 1, \mathcal{F} \right) + \varepsilon L_{\mathbf{los}} L_{\mathbf{hyp}} \cdot T + \sqrt{T \log T}.
$$

*Proof.* This proof is an extension of the proof of Lemma D.5. We note that it suffices to show that the minimax expected regret of the expert $c$ is at most $(M_{\mathcal{P}}(\varepsilon, x_{1:T}) + c + 1) \bar{R}_{\mathcal{B}}^{\mathbf{tr}} \left( \frac{T}{c+1} + 1, \mathcal{F} \right) + \varepsilon L_{\mathbf{los}} L_{\mathbf{hyp}} \cdot T$ for every $c \in [T-1]$, then by the guarantee of MWA (see Theorem C.1), we have our desired upper bound taking $c = \lceil M_{\mathcal{P}}(\varepsilon, x_{1:T}) \rceil$.

Now, we fix a $c \in [T-1]$. Let $\tilde{t}_j = j \left\lceil \frac{T}{c+1} \right\rceil$ for each $j \in [c], \tilde{t}_0 = 0$, and $\tilde{t}_{c+1} = T$. Let $\mathcal{E}$ denote the expert with input $c$ in Algorithm 3, then we have

$$
\mathbb{E}\left[ \sum_{i=1}^{T} \ell(\mathcal{E}(x_t), y_t) \right] = \mathbb{E}\left[ \sum_{j=0}^{c} \sum_{t=\tilde{t}_j+1}^{\tilde{t}_{j+1}} \ell(\mathcal{A}^j(x_t), y_t) \right],
$$

where $\mathcal{A}^j$ is the learner with time duration $[\tilde{t}_j + 1, \tilde{t}_{j+1}]$. Let $m_i$ be the number of mistakes that Predictor $\mathcal{P}$ makes in $\mathcal{A}^j$. Then, by the bound in Lemma D.7, we have the expected loss of $\mathcal{E}$ is at most

$$
\mathbb{E}\left[ \sum_{j=1}^{c} \left( (m_j + 1) \bar{R}_{\mathcal{B}}^{\mathbf{tr}}(\tilde{t}_{j+1} - \tilde{t}_j, \mathcal{F}) + \inf_{f \in \mathcal{F}} \left( \sum_{t=\tilde{t}_j+1}^{\tilde{t}_{j+1}} \ell(f(x_t), y_t) \right) + \varepsilon L_{\mathbf{los}} L_{\mathbf{hyp}} \cdot (\tilde{t}_{j+1} - \tilde{t}_j) \right) \right]
$$

$$
\leq \mathbb{E}\left[ \sum_{j=1}^{c} (m_j + 1) \bar{R}_{\mathcal{B}}^{\mathbf{tr}}(\tilde{t}_{j+1} - \tilde{t}_j, \mathcal{F}) \right] + \inf_{f \in \mathcal{F}} \left( \sum_{t=1}^{T} \ell(f(x_t), y_t) \right) + \varepsilon L_{\mathbf{los}} L_{\mathbf{hyp}} \cdot T.
$$

From the analysis of Lemma D.5, we have

$$\mathbb{E}\left[\sum_{j=1}^{c}(m_j+1)\bar{R}_{\mathcal{B}}^{\mathbf{tr}}(\tilde{t}_{j+1}-\tilde{t}_j,\mathcal{F})\right] \leq (M_{\mathcal{P}}(\varepsilon,x_{1:T})+c+1)\bar{R}_{\mathcal{B}}^{\mathbf{tr}}(\frac{T}{c+1}+1,\mathcal{F}),$$

which proves our desired bound. $\qquad\square$

### D.3  PROOF OF THEOREM 4.2

*Proof.* By Lemma D.5, for any function class $\mathcal{F} \subset \mathcal{Y}^{\mathcal{X}}$, loss function $\ell$, and data stream $(x_{1:T}, y_{1:T})$, the minimax expected regret of Algorithm 4 is bounded by

$$2(M_{\mathcal{P}}(x_{1:T})+1)\bar{R}_{\mathcal{B}}^{\mathbf{tr}}\left(\frac{T}{M_{\mathcal{P}}(x_{1:T})+1}+1,\mathcal{F}\right)+\sqrt{T\log^2 T}.$$

Inputting $M_{\mathcal{P}}(x_{1:T}) = \tilde{\mathcal{O}}\left(T^p\right)$ to the above bound gives us

$$R^{\mathbf{ol}}(T,\mathcal{F})=\tilde{\mathcal{O}}\left(T^p\right)\bar{R}_{\mathcal{B}}^{\mathbf{tr}}\left(T^{1-p},\mathcal{F}\right)+\sqrt{T\log^2 T}.$$

$\qquad\square$

### D.4  PROOF OF COROLLARY 4.3

*Proof.* By Corollary C.9, the minimax expected regret of the transductive online regression for $\mathcal{F}^*$ satisfies $R^{\mathbf{tr}}(T,\mathcal{F}^*)=\tilde{\mathcal{O}}\left(L_{\mathbf{los}}\cdot\sqrt{VT}\right)$. Combining with the bound in Theorem 4.2, we upper bound the minimax expected regret by

$$\tilde{\mathcal{O}}\left(T^p\cdot L_{\mathbf{los}}\cdot\sqrt{VT^{1-p}}\right)+\sqrt{T\log^2 T}=\tilde{\mathcal{O}}\left(L_{\mathbf{los}}\cdot T^{\frac{1+p}{2}}\right),$$

which proves our desired bound. Additionally, we assume that the sequence of examples is predictable in our setting, i.e., $p < 1$. Thus, the minimax expected regret of our algorithm is $o(T)$. $\quad\square$

### D.5  PROOF OF THEOREM 4.4

*Proof.* By Lemma D.8, for any function class $\mathcal{F} \subset \mathcal{Y}^{\mathcal{X}}$ of $L_{\mathbf{hyp}}$-Lipschitz function, $L_{\mathbf{los}}$-Lipschitz and convex loss function $\ell$, and data stream $(x_1,y_1),\ldots,(x_T,y_T)$, the expected loss of Algorithm 6 is bounded by

$$2(M_{\mathcal{P}}(\varepsilon,x_{1:T})+1)\bar{R}_{\mathcal{B}}^{\mathbf{tr}}\left(\frac{T}{M_{\mathcal{P}}(\varepsilon,x_{1:T})+1}+1,\mathcal{F}\right)+\varepsilon L_{\mathbf{los}}L_{\mathbf{hyp}}\cdot T+\sqrt{T\log^2 T}.$$

Inputting $M_{\mathcal{P}}(\varepsilon,x_{1:T})=\tilde{\mathcal{O}}\left(\frac{T^p}{\varepsilon^q}\right)$ to the above bound gives

$$R^{\mathbf{ol}}(T,\mathcal{F})=\inf_{\varepsilon>0}\left\{\tilde{\mathcal{O}}\left(\frac{T^p}{\varepsilon^q}\right)\bar{R}_{\mathcal{B}}^{\mathbf{tr}}\left(\varepsilon^q T^{1-p},\mathcal{F}\right)+\varepsilon L_{\mathbf{los}}L_{\mathbf{hyp}}\cdot T+\sqrt{T\log^2 T}\right\}.$$

Thus, we finish the proof. $\qquad\square$

### D.6  PROOF OF COROLLARY 4.5

*Proof.* By Corollary C.9, the minimax expected regret of the transductive online regression for $\mathcal{F}^*$ satisfies $R^{\mathbf{tr}}(T,\mathcal{F}^*)=\tilde{\mathcal{O}}\left(L_{\mathbf{los}}\cdot\sqrt{VT}\right)$. Combining with the bound in Theorem 4.4, we upper bound the minimax expected regret by

$$\inf_{\varepsilon>0}\left\{\tilde{\mathcal{O}}\left(\frac{T^{\frac{1+p}{2}}}{\varepsilon^{\frac{q}{2}}}\cdot L_{\mathbf{los}}\sqrt{V}\right)+\varepsilon L_{\mathbf{los}}L_{\mathbf{hyp}}\cdot T+\sqrt{T\log^2 T}\right\}=\tilde{\mathcal{O}}\left(L_{\mathbf{los}}L_{\mathbf{hyp}}^{\frac{q}{q+2}}\cdot T^{\frac{p+q+1}{q+2}}\right).$$

Suppose that $L_{\mathbf{hyp}}=\tilde{\mathcal{O}}\left(T^c\right)$ for some constant $c$, then we have $R^{\mathbf{ol}}(T,\mathcal{F})=\tilde{\mathcal{O}}\left(T^{\frac{p+(c+1)q+1}{q+2}}\right)$, so $\mathcal{F}^*$ is learnable if $p+cq<1$. $\qquad\square$

# E  EXPERIMENTS

In this section, we present experiments to justify our theoretical results. The experiments are conducted using an Apple M2 CPU, with 16 GB RAM and 8 cores.

**Experiment Setup.** We generate the sequence of examples $x_{1:T} \subset \mathbb{R}^d$ by a linear dynamical system: $x_{t+1} = Ax_t$, where $A \in \mathbb{R}^{d \times d}$ is a random stable matrix. We restrict each example to having a support size $c < d$. In particular, we randomly select $c$ indices from $[d]$ and set $x_t(i) = 0$ for all other indices and $t \in [T]$. We also set the transition matrix $A$ to have the same support as $x_t$. This construction captures the sparse nature of the sequence of examples in various applications. For instance, in energy management (see Example 1.1), a list of variables $x_t \in \mathbb{R}^d$ can affect the energy consumption $y_t$, however, in specific circumstances, only $c$ of these variables have a significant influence (e.g., the temperature may change by only a few during a month), so the others are set to 0 in $x_t$. In the experiment, we compare the accumulative loss in the transductive online setting, where the learner knows the sequence of examples $x_{1:T}$ in advance, and the online setting, where the learner does not have additional information. We show that the learner achieves better performance in the transductive online setting.

We choose the ground-truth function class $\mathcal{F}$ to be all $c$-junta hyperplanes in $\mathbb{R}^d$ with coefficients in $\{-1 + 0.4 \cdot i, i \in [5]\}$. We consider the regression problem under the additive noise model, i.e., we choose the target function $f^*$ from $\mathcal{F}$ randomly and assign the label $y_t$ to each example $x_t$ by $f^*(x_t) + g_t$, where $g_t \in \mathcal{N}(0, 0.01)$ is random Gaussian noise, representing noisy measurements in real-life scenarios. We evaluate the learner using the $\ell_1$-loss function: $\ell(y, \widehat{y}) = |y - \widehat{y}|$. We choose the parameters $d = 8, c = 4$, and $T = 1000$.

**Methods.** In the experiment, we implement the multiplicative weight algorithm (MWA), which randomly samples the advice of $K$ experts. For the baseline method, we set the experts as the entire function class $\mathcal{F}$, since the learner does not have information on $x_{1:T}$ in the online setting. In contrast, in the transductive online setting, the learner observes $x_{1:T}$ in advance, and so it knows the support $\{i_1, \ldots, i_c\} \subset [d]$ of $x_t$. Thus, we implement MWA on the restricted function class, which is a subset of $\mathcal{F}$ and has $c$-juntas being $\{i_1, \ldots, i_c\}$. We compute the total $\ell_1$-loss of both methods at all times $t \in [T]$. We run 10 repetitions and plot the mean loss curve.

**Results.** As shown in Figure 1, MWA with the restricted net (red line) exhibits a much steeper initial decline in cumulative loss compared to MWA on the entire net (blue line), indicating faster convergence toward the ground-truth function. In addition, the restricted net consistently maintains a lower error throughout all rounds, with the gap widening over time.

The experiment demonstrates that using the support information of the input sequence of examples significantly improves learning outcomes. The result highlights the empirical separation between the standard online setting and the transductive online setting: access to the example sequence allows the learner to focus on a refined function class, thereby achieving lower regret in practice.

1728
1729
1730
1731
1732
1733
1734
1735
1736
1737
1738
1739
1740
1741
1742
1743
1744
1745
1746
1747
1748
1749
1750
1751
1752
1753
1754
1755
1756
1757
1758
1759
1760
1761
1762
1763
1764
1765
1766
1767
1768
1769
1770
1771
1772
1773
1774
1775
1776
1777
1778
1779
1780
1781

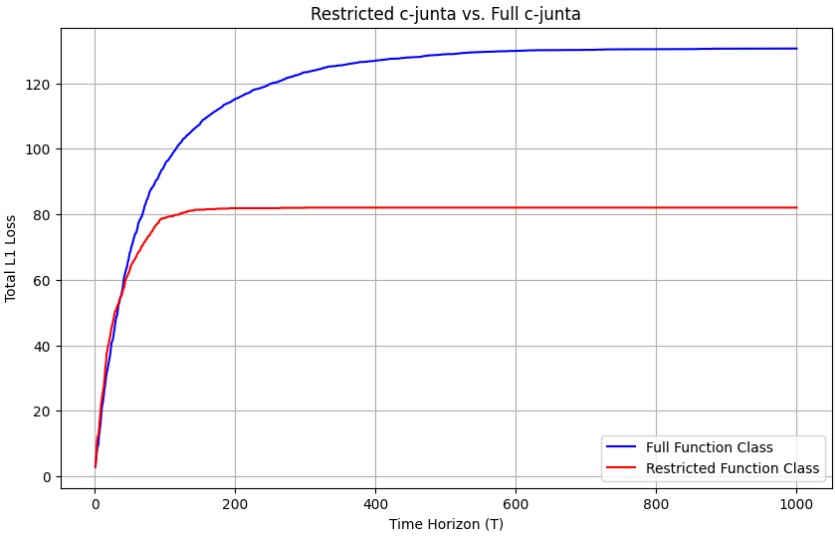

Fig. 1: Comparison between the performance of MWA on the entire net and the restricted net. The blue line is the entire net, and the red line is the restricted net.

