# OpenReview forum: "Transductive and Learning-Augmented Online Regression"
_ICLR.cc/2026/Conference — ICLR 2026 Conference Withdrawn Submission_

### Official Review · Reviewer_JgLE · 2025-10-27

**Soundness:** 3
**Presentation:** 3
**Contribution:** 2
**Rating:** 4
**Confidence:** 3

**Summary:**

This paper studies online learning under two related settings: the *transductive* setting, where the entire sequence of features or contexts is revealed to the learner in advance, and the *learning-augmented* setting, where the learner has access to a predictor that provides (possibly noisy) forecasts of future examples. The authors analyze two types of prediction errors—the zero-one error and the $\epsilon$-ball error defined with respect to a metric on the context space. The main contributions are a minimax-optimal bound for the transductive setting and upper bounds for online learning with imperfect predictions, where the latter scale with the transductive regret multiplied by a term that depends on the predictor’s error.

**Strengths:**

1. The main contribution, as I understand it, is the learning-augmented setting, which extends the classification scenario from Raman & Tewari (2024) to regression.

2. The paper introduces some new techniques for handling the regression case and connects it to the concept of transductive regret.

**Weaknesses:**

1. I have a major concern regarding the claimed novelty of the minimax regret result and its connection to the fat-shattering dimension in transductive regression. This connection, in my understanding, was already established in Lemma 11 and Lemma 12 of A. Rakhlin, O. Shamir, and K. Sridharan, *“Relax and Randomize: From Value to Algorithms,”* NeurIPS 2012. Their work derived a minimax-optimal regret bound for the transductive setting expressed via the (classical) Rademacher complexity, and it is a standard argument to convert such Rademacher-based bounds into equivalent statements in terms of the fat-shattering dimension, as done in the present paper. Moreover, Rakhlin et al. not only established this characterization but also provided an oracle-efficient algorithm achieving the optimal regret, whereas the current submission only re-derives the bound without offering a constructive realization.

2. Given point 1 above, the actual contribution of the paper lies in the learning-augmented setting, where the authors establish only an upper bound. However, it remains unclear how tight this bound is with respect to both the prediction error and the transductive regret. Moreover, the overall results seems (conceptually) largely follows Raman & Tewari (2024).

**Questions:**

Could you clarify the claimed novelty of your transductive minimax regret result?

---

### Official Review · Reviewer_AyX2 · 2025-10-30

**Soundness:** 2
**Presentation:** 3
**Contribution:** 2
**Rating:** 4
**Confidence:** 3

**Summary:**

This paper investigates the online regression in both the transductive and learning augmented settings.  In the transductive setting, it establishes the near-optimal regret bound for $\ell_1$ loss. In the online regression with predictions setting, it proposes the algorithms that adapt smoothly to the quality of the Predictor. They also conduct numerical experiments to verify the effectiveness of their methods.

**Strengths:**

* The paper is well organized and easy to follow. The authors also provide several applications of the online regression setting.
* The theoretical results resolve an open problem left by previous studies.
* The work presents a unified and elegant theoretical framework.
* The paper also includes numerical experiments to support the theoretical claims.

**Weaknesses:**

**W1.** I am confused about the motivation of this paper, especially in Lines 80–83. Why do the authors investigate the use of future examples to obtain better-than-worst-case regret bounds?

**W2.** In Line 131, the authors note that the setting where the learner has access to samples is unrealistic. Therefore, I wonder whether the contribution of the first results is actually limited.

**W3.** What is the valid range of $p$ in Corollary 1.3?

**W4.** The equation in Lines 252–255 exceeds the column width.

**W5.** The algorithm design and analysis of this paper are largely based on Raman & Tewari (2024), so I have serious concerns about the novelty of this work. I hope the authors can highlight what specific challenges arise in the online regression setting compared with prior work.

**W6.** Can the authors conduct experiments on real-world datasets to further validate their method?

**W7.** The authors place the pseudocode of the algorithm in the appendix, but I believe it should be included in the main text for better readability.

**Questions:**

Refer to __Weakness__

---

### Official Review · Reviewer_RXvk · 2025-11-02

**Soundness:** 3
**Presentation:** 3
**Contribution:** 2
**Rating:** 6
**Confidence:** 3

**Summary:**

The paper studies online regression when the learner has access to predictions (ranging from full knowledge of the unlabeled input sequence to noisy/evolving forecasts). It proves that in the transductive setting the minimax expected regret is governed by the fat shattering dimension, allowing learning and yielding tighter rates for classes like bounded variation. It then designs learning augmented algorithms with restarts and multiplicative weights over interval granularities so that regret interpolates between worst case and transductive bounds as predictor quality improves. A small synthetic experiment qualitatively illustrates the gap.

**Strengths:**

- Clear theoretical separation: transductive regret depends on fat shattering (not sequential), enabling new learnability results for key classes
- Interpolation guarantees with concrete mistake models (0-1 and eps-ball) and constructive algorithms (restarts + MW over interval counts)
- Useful instantiations (Lipschitz, k-fold aggregations, bounded variation) with near-matching bounds that make the results tangible

**Weaknesses:**

- Practicality/constructiveness: the main transductive upper bound is largely minimax style; the explicit MWA-based learner is suboptimal, and computational aspects (cover construction, oracle assumptions, runtime/memory) are under-specified. It'd be useful to clarify its feasibility and provide a practical surrogate.
- Empirics are thin: only a single synthetic study; no evaluation that varies predictor quality M or eps, or uses no real data streams, or compares to strong online baselines with imperfect forecasts. A more systematic experimental section would greatly strengthen the paper.

**Questions:**

Please address the concerns raised above.

---

### Official Review · Reviewer_3U75 · 2025-11-02

**Soundness:** 3
**Presentation:** 3
**Contribution:** 2
**Rating:** 6
**Confidence:** 4

**Summary:**

This paper studies online regression beyond the standard adversarial setting. In particular, the authors provide results for two settings:
- The transductive setting, where the sequence of examples (but not their corresponding labels) is revealed up front,
- The prediction setting, where the learner has access to a predictor of the sequence of examples, and its performance depends on the accuracy of such predictors.

For the transductive setting, the authors prove that the combinatorial parameter underlying learnability is the fat-shattering dimension, which is strictly smaller than the sequential fat-shattering dimension, which characterizes online learnability in general. This result is existential, as it follows from a minimax argument. The authors also provide a simple learning algorithm (compute an eps-cover of the examples in the sequence and run MWU on them) with a suboptimal rate.

The authors also provide positive results for the learning augmented setting, where the learner receives predictions on the examples in advance. Here, two metrics of the predictor sub-routines are used: the $0/1$ loss, and the $\varepsilon$-ball loss.

**Strengths:**

- Online Regression is a classical topic with vast applicability.
- The theoretical results are sound and complete. In particular, the general result for transductive learning is nearly tight.
- The main body is generally well written.

**Weaknesses:**

- The regret rate for the transductive setting is only existential. This fact should be stressed earlier in the paper, at least in the introduction. It is now only mentioned in Section 3.
- I find it challenging to assess the novelty and technical contribution of the paper:
- The transductive result seems to follow easily from the adversarial analysis, replacing the sequential Rademacher complexity with its non-sequential version, and using that the sequence of examples is known in advance. The practical algorithm for that setting is also fairly simple.
- The algorithm for the 0/1 loss (which is then also used for the eps-ball loss) seems to heavily rely on Raman & Tewari.

Minor Comments:
- line 130 “online learning” -> “online learnable”
- line 159: L_{loss} is defined a couple of pages later
- The bibliography is disorganized: multiple styles are used. In particular, the NeurIPS 23 paper by Hanneke et al is cited in two different entries, the JMLR 15 by Rakhlin et al is also cited twice, while the citation in lines 684 and 685 is missing the name of the journal/conference.

**Questions:**

Could you please elaborate on the novelty and the technical contributions?

---

### Note · Authors · 2025-12-03

I have read and agree with the venue's withdrawal policy on behalf of myself and my co-authors.